# Single-cell sequencing provides clues about the developmental genetic basis of evolutionary adaptations in syngnathid fishes

Hope M Healey[1,2]*, Hayden B Penn[1], Clayton M Small[1,3], Susan Bassham[1], Vithika Goyal[1], Micah A Woods[1], William A Cresko[1,2]*

[1]Institute of Ecology and Evolution, University of Oregon, Eugene, United States; [2]Knight Campus for Accelerating Scientific Impact, University of Oregon, Eugene, United States; [3]School of Computer and Data Science, University of Oregon, Eugene, United States

*For correspondence:
hhealey@uoregon.edu (HMH);
wcresko@uoregon.edu (WAC)

**Competing interest:** The authors declare that no competing interests exist.

## eLife Assessment

This study provides a **valuable** new resource to investigate the molecular basis of the particular features characterizing the pipefish embryo. The authors found both unique and shared gene expression patterns in pipefish organs compared with other teleost fishes. The **solid** data collected in this unconventional model organism will give new insights into understanding the extraordinary adaptations of the Syngnathidae family and will be of interest in the domain of evolution of fish development.

**Abstract** Seahorses, pipefishes, and seadragons are fishes from the family Syngnathidae that have evolved extraordinary traits including male pregnancy, elongated snouts, loss of teeth, and dermal bony armor. The developmental genetic and cellular changes that led to the evolution of these traits are largely unknown. Recent syngnathid genome assemblies revealed suggestive gene content differences and provided the opportunity for detailed genetic analyses. We created a single-cell RNA sequencing atlas of Gulf pipefish embryos to understand the developmental basis of four traits: derived head shape, toothlessness, dermal armor, and male pregnancy. We completed marker gene analyses, built genetic networks, and examined the spatial expression of select genes. We identified osteochondrogenic mesenchymal cells in the elongating face that express regulatory genes *bmp4*, *sfrp1a*, and *prdm16*. We found no evidence for tooth primordia cells, and we observed re-deployment of osteoblast genetic networks in developing dermal armor. Finally, we found that epidermal cells expressed nutrient processing and environmental sensing genes, potentially relevant for the brooding environment. The examined pipefish evolutionary innovations are composed of recognizable cell types, suggesting that derived features originate from changes within existing gene networks. Future work addressing syngnathid gene networks across multiple stages and species is essential for understanding how the novelties of these fish evolved.

## Introduction

Seahorses, pipefishes, and seadragons are extraordinary fishes in the family Syngnathidae with diverse body plans, coloration, and elaborate structures for paternal brooding. The syngnathid clade comprises over 300 diverse species that vary in conservation status, distribution, ecology, and

**Figure 1.** Gulf pipefish exemplify syngnathid-derived traits. Gulf pipefish have elongate snouts, have lost teeth on their oral and pharyngeal jaws, possess dermal armor, and have brood pouches in males (**A, B**). Cartilage (alcian in blue) and bone (alizarin in red) stained clutch siblings of embryos from the two single-cell RNA sequencing (scRNAseq) samples are shown in **C-H**. Embryos have cartilaginous craniofacial skeletons (**B, C, F; E** marks the (Mes)ethmoid cartilage, **C** indicates the Ceratohyal, **H** shows the Hyosymplectic cartilage, **M** marks the Meckel's cartilage, **Q** indicates the quadrate, **B** shows the Basihyal, and **P** marks the palatoquadrate) with the onset of ossification in the jaw. They have cartilaginous fin radials in the dorsal fin (**D, G; F** indicates fin radials and **N** denotes the notochord). The embryos do not have signs of ossification in the trunk where the exoskeleton will form later (**E, H**; PD show the region where dermal armor primordia will arise). Panels **C, D, F, and G**, scale bar 200 µm; **E, H** scale bar 100 µm.

morphology (*Leysen et al., 2011*; *Manning et al., 2019*; *Schneider et al., 2023b*; *Stiller et al., 2022*). Syngnathids have numerous highly altered traits, trait losses, and evolutionary novelties. They have elongated snouts bearing small, toothless jaws (*Leysen et al., 2011*) specialized for the capture of small zooplankton (*Van Wassenbergh et al., 2009*; *Flammang et al., 2009*; *Bergert and Wainwright, 1997*). Additionally, syngnathids are distinctly protected by a bony dermal armor rather than scales (*Jungerson, 1910*). Other skeletal differences include a lack of ribs and pelvic fins, and an expansion in the number of vertebrae (*Schneider et al., 2023b*). Finally, syngnathids exhibit male pregnancy, which has involved the evolution of specialized brooding tissues and structures (*Whittington and Friesen, 2020*). Paternal investment varies among lineages; for example, seadragons tether embryos externally to their tails while seahorses and some pipefishes have enclosed brood pouches proposed to support embryos through nutrient transfer and osmotic regulation (*Carcupino, 2002*; *Melamed et al., 2005*; *Ripley and Foran, 2006*).

Despite advances in understanding the ecology and evolution of syngnathid novelties, the developmental genetic basis for these traits is largely unknown. The recent production of high-quality syngnathid genome assemblies (*Qu et al., 2021*; *Ramesh et al., 2023*; *Small et al., 2022*; *Wolf et al., 2024*) provides initial clues for the developmental genetic basis of some evolutionary changes. Studies have found that syngnathids lack several genes with deeply conserved roles in vertebrate development, including pharyngeal arch development (*fgf3*), tooth development (*fgf3*, *fgf4*, and *eve1*, and most Scpp genes), fin development (*tbx4*), and immune function (MHC pathway components) (*Lin et al., 2016*; *Qu et al., 2021*; *Small et al., 2016*; *Small et al., 2022*; *Zhang et al., 2020*). Though these gene losses are highly suggestive of leading to unique changes, exploration of the actual developmental consequences of their losses is needed.

To fill this gap in knowledge, we used single-cell RNA sequencing (scRNAseq) to investigate how these striking genomic changes have affected the developmental genetic and cellular basis of syngnathids' derived traits. The Gulf pipefish (*Syngnathus scovelli*) is an attractive model for this study (*Figure 1A*). This species has a high-quality reference genome annotated by NCBI and is amenable to laboratory culture (*Anderson and Jones, 2019*; *Ramesh et al., 2023*). Furthermore, species from the *Syngnathus* genus are used worldwide to address questions about syngnathid evolution in microbial, developmental, functional morphological, histological, transcriptomic, ecotoxicological, and genomic studies (*Berglund et al., 1986*; *Carcupino, 2002*; *Fuiten and Cresko, 2021*; *Harada et al., 2022*; *Harlin-Cognato et al., 2006*; *Partridge et al., 2007*; *Ripley and Foran, 2006*; *Rose et al., 2023*; *Roth et al., 2012*; *Small et al., 2016*; *Small et al., 2013*).

In this paper, we focus on a subset of unique syngnathid traits, including their elongated head, toothlessness, dermal armor, and development of embryos inside the brood pouch. These traits represent the diversity of evolutionary changes observed in the syngnathid clade (highly altered, lost, and novel traits), and hypotheses from studies of model organisms suggest developmental pathways involved in their evolution (*Lin et al., 2016*; *Roth et al., 2020*; *Small et al., 2016*; *Small et al., 2022*).

scRNAseq atlases are a powerful complement to genomic analyses (*Shema et al., 2019*; *Ton et al., 2020*). They can provide crucial insights into the types of cells present, genes that distinguish cell types (marker genes), active gene networks, and a means to identify the expression of genes of interest within predicted cell types (*Farnsworth et al., 2020*; *Farrell et al., 2018*; *Williams et al., 2019*). Specifically, scRNAseq captures RNA expression profiles from individual cells, allowing cell types to be inferred *post-hoc.* scRNAseq has successfully been applied to syngnathid adult kidneys (*Parker et al., 2022*), but there are no published syngnathid developmental atlases.

Here, we report the first developmental scRNAseq atlas for syngnathids from late embryogenesis staged Gulf pipefish. We delineate the overall structure of this atlas, which describes 38 cell clusters composed of 35,785 cells, and use these data to make inferences about the morphological evolution of several syngnathid innovations. In addition to inferring present cell types and their underlying genetic networks, we detail spatial expression patterns using *in situ* hybridization experiments of select markers and other candidate genes in pipefish embryos and juveniles. We found genes of conserved signaling pathways expressed during craniofacial development but did not detect evidence of tooth primordia. The embryonic dermis and epidermis, respectively, expressed genes for the dermal armor development (bone development pathways) and genes potentially involved in interaction with the male brood pouch (e.g. nutrient acquisition genes). Overall, this atlas provides a deeper understanding of the development of Gulf pipefish and identifies gene candidates for understanding the development of syngnathid evolutionary innovations. In addition to these discoveries, this atlas provides a significant resource for researchers studying syngnathid evolution and development.

## Results

### Valuable scRNAseq atlas for studying syngnathid development

We produced the first developmental scRNAseq atlas for a syngnathid from two samples comprising 20 similarly staged embryos from pregnant, wild-caught Gulf pipefish (*Syngnathus scovelli*) males. The samples represent a late organogenesis developmental stage (*Figure 1B–H*). These embryos had a primarily cartilaginous skeleton with minimal mineralization, including jaw cartilages that were at the onset of mineralization and ethmoid elongation. The embryos also possessed cartilaginous dorsal fin pterygiophores but had no signs of dermal armor mineralization. This stage is referred to as 'frontal jaws' in the literature on syngnathids (*Sommer et al., 2012*).

The atlas included 35,785 cells (19,892 and 15,893 cells from each sample; *Figure 2—figure supplement 1*; *Figure 2—figure supplement 2*), which formed 38 cell clusters (*Figure 2A*, *Supplementary file 1*; *Supplementary file 2*). We classified cells into four different broad tissue types – epithelial, connective, neural, and muscle – using Seurat-identified marker genes and published model organism resources (*Figure 2—figure supplements 3–58*). We next used Seurat-identified marker genes to pinpoint single marker genes that were most unique to each cluster (*Figure 2—figure supplement 59*). We completed *in situ* hybridization using Gulf and bay pipefish embryos for cell clusters for which examining gene expression would help hone and validate cluster annotations (*Figure 2—figure supplements 60–72*).

In total, our atlas contained 13,027 connective tissue cells (excluding cells from the blood, immune, and the digestive system) from 14 clusters, 10,112 nervous system cells from 10 clusters, 4,363 muscle cells from five clusters, 4,133 blood cells from three clusters, 650 immune cells from two clusters, 432 pigment cells from one cluster, 370 epidermal cells from one cluster, and 137 gut cells from one cluster. Within the connective tissue cell types, we also identified cartilage (302 cells), developing bone (442 cells), fins (253 cells), and notochord (693 cells). The number of recovered cells per identity may not necessarily represent organismal cellular proportions because of potential

variability in dissociation success for different cell types (*Denisenko et al., 2020*; *Uniken Venema et al., 2022*).

## Discovery of cell cluster function and state using KEGG analysis

To affirm identities and discover the potential properties of each cluster, we completed a KEGG pathway analysis for each cluster using Seurat's marker genes (*Figure 2—figure supplement 73*, *Figure 2B*). For eight of the clusters (1, 4, 6, 9, 11, 15, 19, and 24), we did not find any significantly enriched pathways, possibly due to similar gene expression profiles across cell types that reduced the number of identified markers. However, we found one or more significantly enriched pathways for the other 29 cell clusters. We observed enriched pathway terms that supported cluster annotations. For example, 'phototransduction' in the retina cluster, 'melanogenesis' in the pigment cluster, 'cardiac muscle contraction' in muscle clusters, and 'neuroactive ligand-receptor interaction' in neuronal clusters.

The inferred KEGG pathways demonstrated some commonalities across the different tissue types, including in signaling pathways and cell states. Notably, our identified KEGG terms delineated progenitor and differentiated cell clusters. Based on their KEGG terms, we classified clusters 8, 10, and 16 as possible neural, muscle, and connective tissue progenitor cells, respectively. We also detected expression of *pax3a* and *pax3b*, muscle primordia markers, in cluster 10, supporting this annotation. These clusters had enriched KEGG terms associated with cell division ('cell cycle,' 'DNA replication,' 'nucleotide excision repair,' and 'homologous recombination'), and lacked enrichment for KEGG pathways present with differentiated cell types of their lineage. Specifically, cluster 8 lacked the neural KEGG term 'neuroactive ligand-receptor interaction,' cluster 10 lacked muscle KEGG terms 'adrenergic signaling in cardiomyocytes,' 'calcium signaling pathways,' and 'cardiac muscle contraction,' and cluster 16 lacked connective tissue term 'ECM receptor interaction.' To complement these findings, we completed a cell differentiation analysis using CytoTRACE for neural, muscle, and connective clusters (*Figure 2—figure supplement 74*). Clusters 8, 10, and 16 had the lowest scores in each respective comparison, which indicated undifferentiated cell states. Thus, it is likely that clusters 8, 10, and 16 represented undifferentiated cells within the major lineages of neural, muscle, and connective cells.

## Commonalities of cell clusters, unique networks, and elusive cell types identified in network analysis

We built gene networks/modules from 3000 variable genes using weighted gene network correlation analysis (WGCNA; *Langfelder and Horvath, 2008*). This produced 43 gene modules in total (*Supplementary file 8*; *Supplementary file 9*), assessed for each cluster-module pair for their strength of association (*Figure 3A*, *Supplementary file 10*; *Supplementary file 11*) and every module's dependence on each cluster for their network connectivity (*Figure 3—figure supplement 1*, *Supplementary file 12*; *Supplementary file 13*). Using the genes from each network, we completed a KEGG pathway analysis to identify whether gene modules indicated specific cellular pathways or states (*Figure 3B*). We initially explored whether these network-cluster associations could reveal commonalities between cell clusters or identify whether particular clusters contained multiple cell identities.

First, we asked whether gene modules that associate with three or more cell clusters signify commonalities between clusters that have similar cell types. We identified seven gene modules (6, 7, 21, 14, 17, 41, and 42) that each associate with three or more cell clusters. These gene modules do connect clusters of similar identities or cell states. For example, Modules 6 and 7 are associate with connective tissue cells. Module 6, the larger of the two modules, contains numerous other KEGG pathways found in most connective cell clusters, such as 'cytokine-cytokine receptor interaction' (*Figure 3C*). Interestingly, module 21, associated with clusters 8, 10, 16, and 25, contains genes from KEGG pathways related to the 'cell cycle' and 'cellular senescence', supporting the results found in our cluster-based KEGG pathway analysis (*Figure 3D*). Module 21 also contained 'Notch signaling' genes, possibly due to similar correlations with cell cycle genes; however, these were only expressed in the neural cell clusters (8 and 25).

Where one cell cluster is associated with multiple gene networks, we wondered if multiple cell identities existed within the cell cluster. We explored this possibility by examining the pigment cell

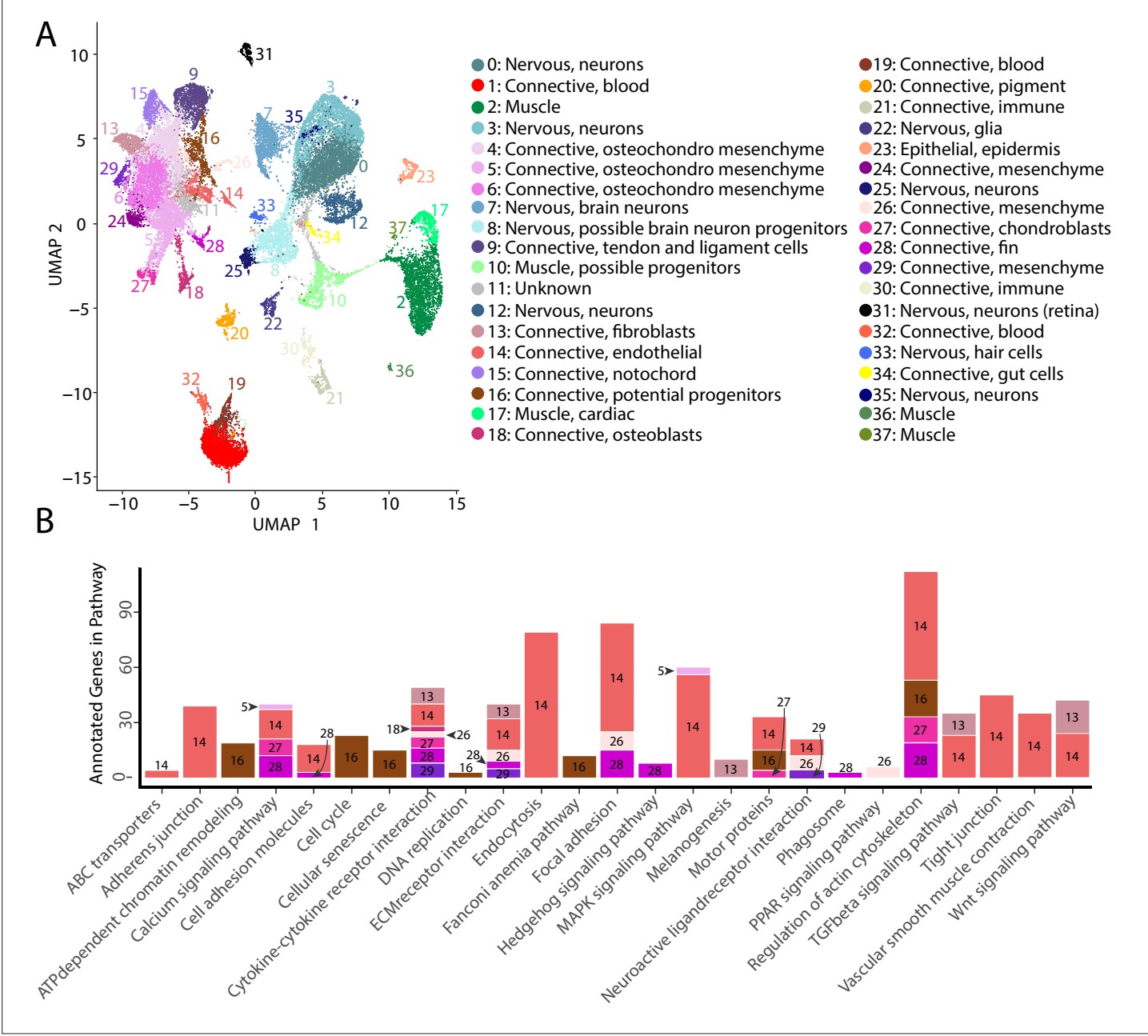

**Figure 2.** Gulf pipefish single-cell atlas contains cells from the entire embryo and identifies genetic pathways active in different cell types. The UMAP plot (**A**) shows all of the cell clusters and their identities reduced to the first two UMAP dimensions. The graph in (**B**) displays the results of the KEGG pathway analysis in cell clusters identified as connective tissue (excluding blood, pigment, digestive, and immune cells). The number of Seurat-identified marker genes for each cluster that was a part of each pathway is displayed on the y-axis. Bars are colored and labeled by cell cluster.

The online version of this article includes the following figure supplement(s) for figure 2:

**Figure supplement 1.** Two Gulf pipefish samples have similar contributions to cell clusters.

**Figure supplement 2.** QC metrics for both Gulf pipefish samples show similar sample quality after initial filtering.

**Figure supplement 3.** Zebrafish muscle cell marker genes expression patterns correspond with annotated muscle cells in the Gulf pipefish atlas.

**Figure supplement 4.** Zebrafish muscle satellite cell marker genes expression patterns correspond with annotated muscle primordia cells in the Gulf pipefish atlas.

**Figure supplement 5.** Zebrafish cardiac muscle cell marker gene expression patterns correspond with annotated cardiac muscle cells in the Gulf pipefish atlas.

**Figure supplement 6.** Zebrafish retinal cell marker gene expression patterns correspond with annotated retinal cells in the Gulf pipefish atlas.

*Figure 2 continued on next page*

*Figure 2 continued*

**Figure supplement 7.** Zebrafish neuronal cell marker genes expression patterns correspond with annotated neuronal cells in the Gulf pipefish atlas.

**Figure supplement 8.** Zebrafish chondroblast cell marker gene expression patterns correspond with annotated chondroblast cells in the Gulf pipefish atlas.

**Figure supplement 9.** Zebrafish osteoblast cell marker gene expression patterns correspond with annotated osteoblast cells in Gulf pipefish atlas.

**Figure supplement 10.** Zebrafish notochord cell marker gene expression patterns correspond with annotated notochord cells in the Gulf pipefish atlas.

**Figure supplement 11.** Zebrafish fin cell marker gene expression patterns correspond with annotated fin cells in the Gulf pipefish atlas.

**Figure supplement 12.** Zebrafish mesenchymal cell marker gene expression patterns correspond with annotated mesenchymal cells in the Gulf pipefish atlas.

**Figure supplement 13.** Zebrafish tenocyte cell marker gene expression patterns correspond with annotated tenocyte cells in the Gulf pipefish atlas.

**Figure supplement 14.** Zebrafish erythrocyte cell marker gene expression patterns correspond with annotated erythrocyte cells in the Gulf pipefish atlas.

**Figure supplement 15.** Zebrafish endothelial cell marker gene expression patterns correspond with annotated endothelial cells in the Gulf pipefish atlas.

**Figure supplement 16.** Zebrafish gut and liver cell marker gene expression patterns correspond with annotated gut and liver cells in the Gulf pipefish atlas.

**Figure supplement 17.** Zebrafish epidermal cell marker gene expression patterns correspond with annotated epidermal cells in the Gulf pipefish atlas.

**Figure supplement 18.** Zebrafish glial cell marker gene expression patterns correspond with annotated glial cells in the Gulf pipefish atlas.

**Figure supplement 19.** Zebrafish immune cell marker gene expression patterns correspond with annotated immune cells in the Gulf pipefish atlas.

**Figure supplement 20.** Zebrafish pigment cell marker gene expression patterns correspond with annotated pigment cells in the Gulf pipefish atlas.

**Figure supplement 21.** Zebrafish fibroblast cell marker gene expression patterns correspond with annotated fibroblast cells in the Gulf pipefish atlas.

**Figure supplement 22.** Gulf pipefish cluster 0 markers.

**Figure supplement 23.** Gulf pipefish cluster 1 markers.

**Figure supplement 24.** Gulf pipefish cluster 2 markers.

**Figure supplement 25.** Gulf pipefish cluster 3 markers.

**Figure supplement 26.** Gulf pipefish cluster 4 markers.

**Figure supplement 27.** Gulf pipefish cluster 5 markers.

**Figure supplement 28.** Gulf pipefish cluster 6 markers.

**Figure supplement 29.** Gulf pipefish cluster 7 markers.

**Figure supplement 30.** Gulf pipefish cluster 8 markers.

**Figure supplement 31.** Gulf pipefish cluster 9 markers.

**Figure supplement 32.** Gulf pipefish cluster 10 markers.

**Figure supplement 33.** Gulf pipefish cluster 12 markers.

**Figure supplement 34.** Gulf pipefish cluster 13 markers.

**Figure supplement 35.** Gulf pipefish cluster 14 markers.

**Figure supplement 36.** Gulf pipefish cluster 15 markers.

**Figure supplement 37.** Gulf pipefish cluster 16 markers.

**Figure supplement 38.** Gulf pipefish cluster 17 markers.

**Figure supplement 39.** Gulf pipefish cluster 18 markers.

**Figure supplement 40.** Gulf pipefish cluster 19 markers.

**Figure supplement 41.** Gulf pipefish cluster 20 markers.

**Figure supplement 42.** Gulf pipefish cluster 21 markers.

**Figure supplement 43.** Gulf pipefish cluster 22 markers.

**Figure supplement 44.** Gulf pipefish cluster 23 markers.

**Figure supplement 45.** Gulf pipefish cluster 24 markers.

**Figure supplement 46.** Gulf pipefish cluster 25 markers.

**Figure supplement 47.** Gulf pipefish cluster 26 markers.

**Figure supplement 48.** Gulf pipefish cluster 27 markers.

*Figure 2 continued*

**Figure supplement 59.** Marker genes define cell identity and are a resource for cell cluster exploration.

**Figure supplement 60.** *slc25a4*, a marker for cluster 2, is expressed in muscle cells in bay pipefish.

**Figure supplement 61.** *scg2a*, a marker for cluster 3, is expressed in the brain of bay pipefish.

**Figure supplement 62.** *prdm16*, a marker for cluster 5, is expressed in the head and fins of bay pipefish.

**Figure supplement 63.** *elnb*, a marker for cluster 6, is expressed in the head of a bay pipefish.

**Figure supplement 64.** *fndc9*, a marker for cluster 7, is expressed in the brain of Gulf pipefish.

**Figure supplement 65.** *insm1b*, a marker for cluster 8, is expressed in the brain of Gulf pipefish.

**Figure supplement 66.** *tnmd*, a marker for cluster 9, is expressed in tendons and ligaments of bay pipefish.

**Figure supplement 67.** *myf5*, a marker for cluster 10, is expressed in the muscle of bay pipefish.

**Figure supplement 68.** *hpdb*, a marker for cluster 13, is expressed in somites of bay pipefish.

**Figure supplement 69.** *ifitm5*, a marker for cluster 18, is expressed in bay pipefish osteoblasts.

**Figure supplement 70.** *plaat4*, a marker for cluster 23, is weakly expressed in bay pipefish epidermal cells.

**Figure supplement 71.** *hyal6*, a marker for cluster 33, is strongly expressed in bay pipefish hair cells.

**Figure supplement 72.** *soat2*, a marker for cluster 36, is strongly expressed in bay pipefish muscle cells.

**Figure supplement 73.** KEGG analysis reveals pathways significantly enriched in cell clusters.

**Figure supplement 74.** Clusters 8,10, and 16 are composed of largely undifferentiated cells.

cluster (cluster #20) and whether the five different correlated gene modules (#34, 35, 36, 37, and 38) are expressed in distinct subgroups of cells within the cluster (*Figure 3—figure supplement 1*). We found cases of more than one genetic network (modules 35 and 38) expressed in the same cells. Modules 35 and 38 included conserved pigment genes, *pmela*, *mlana*, and *dct* in module 35 and *tryp1b* and *pmel* in module 38, that mark melanocytes (*Figure 3—figure supplement 2*; *Du et al., 2003*; *Johnson et al., 2011*; *Lamason et al., 2005*; *Thisse and Thisse, 2004*). However, we also found non-overlapping expression of networks, notably modules 36 and 37. We inferred that module 36 is associated with xanthophores and xanthoblasts due to the presence of *plin6* and *scarb1*, genes involved with lipid binding and activity in xanthophores (*Ahi et al., 2020*). On the other hand, module 37 likely represents iridophores and iridoblasts because it contains *pnp4a*, which is involved in purine-nucleoside phosphorylase activity in iridoblasts (*Kimura et al., 2017*).

## Conserved signaling pathways are active during syngnathid craniofacial development

The specialized pipefish feeding apparatus is composed of an elongate, tubular snout, toothless mandible and pharyngeal jaws, large tendons, and associated muscles. Therefore, numerous cell types contribute to their distinct faces: cartilage, bone, tendon, muscle, and connective tissues as well as their progenitors. We sought to identify markers of these cell types and the signaling pathways active in them.

We found marker genes uniquely expressed in the face, genes that mark cell types important to craniofacial development, and markers with potentially relevant functions for craniofacial development using *in situ* hybridizations of cell cluster marker genes. For instance, we observed the marker for

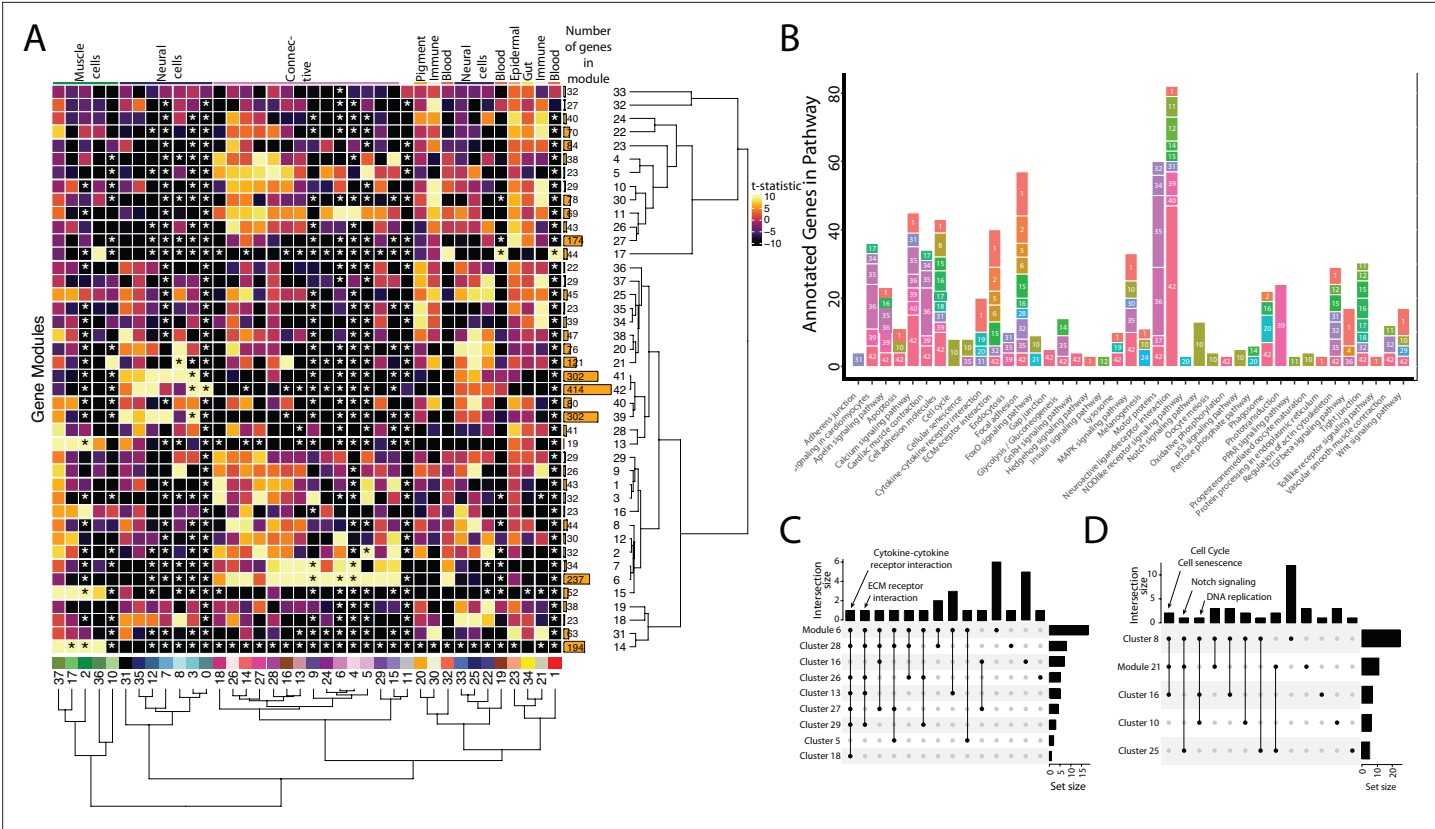

**Figure 3.** Weighted Gene Network Analysis (WGCNA) identifies gene modules that define and unite cell clusters. (**A**) The strength of association between the gene modules and cell clusters is shown in panel A with dendrogram clustering illustrating the distance between modules and cell clusters. Gene modules are represented by rows and cell clusters by columns. The modules and clusters are clustered using the Pearson distance method. The number of genes in each gene module are shown in the right-hand bar plots. Cell clusters are colored based on their identity. The asterisks indicate the module-cluster relationships that have a p-value less than 0.05 from a two-sided permutation test after correction for multiple tests (false discovery rate, FDR). The heatmap is colored by t-statistics in a range of –10 to 10, with highly positive values in yellow and highly negative values in black. (**B**) The identified gene modules possess genes from KEGG pathways. The bars are labeled with the gene module and the size of each bar corresponds to the number of genes from the KEGG pathway in the module. Since WGCNA modules do not have p-values, only KEGG pathways with more than two genes included in the gene module are shown on the plot. (**C**) Identified gene modules contain similar KEGG pathways as the cell clusters that correlated with them. These relationships are shown in Upset plots where each row is a cell cluster or gene module, each column represents KEGG pathways shared by the modules and clusters (shared condition is shown filled in black dots connected by lines), the interaction size is the number of pathways in common between the set of modules and clusters, and the set size is the number of pathways that are enriched in each cluster and module. Panel C1 highlights that 'cytokine-cytokine receptor interaction' and 'ECM receptor interaction' are present in module 6 as well as 6 and 4 connective cell clusters, respectively. Panel C2 shows that 'cell cycle' and 'senescence' are present in module 21 as well as clusters 8 and 16, 'Notch signaling' genes are present in clusters 8 and 25 as well as module 21, and 'DNA replication' is present in clusters 8, 10, and 16.

The online version of this article includes the following figure supplement(s) for figure 3:

**Figure supplement 1.** Cell clusters drive gene module connectivity.

**Figure supplement 2.** Gene module expression comparison with the Pigment Cluster reveals potential elusive cell types within the cluster.

osteochondro-mesenchymal cells (cluster #6), *elnb*, specifically expressed at the intersection between the ethmoid plate and palatoquadrate as well as on the Meckel's cartilage (***Figure 4B***). Although *elnb* is observed in the zebrafish cranial skeleton, it is primarily studied for its proposed role in teleost heart evolution (***Miao et al., 2007***; ***Moriyama et al., 2016***).

Other genes identified here as cell markers were not uniquely craniofacial but provided insights into the cell types that comprise the face. For example, *tnmd* expression marked tendons and ligaments (cluster #9) throughout the face and body (***Figure 4C***; ***Figure 2—figure supplement 67***). Our finding is consistent with *tnmd*'s role in tenocyte development in model systems, namely zebrafish and mouse (***Chen and Galloway, 2014***; ***Docheva et al., 2005***). Identifying tenocyte cells is particularly

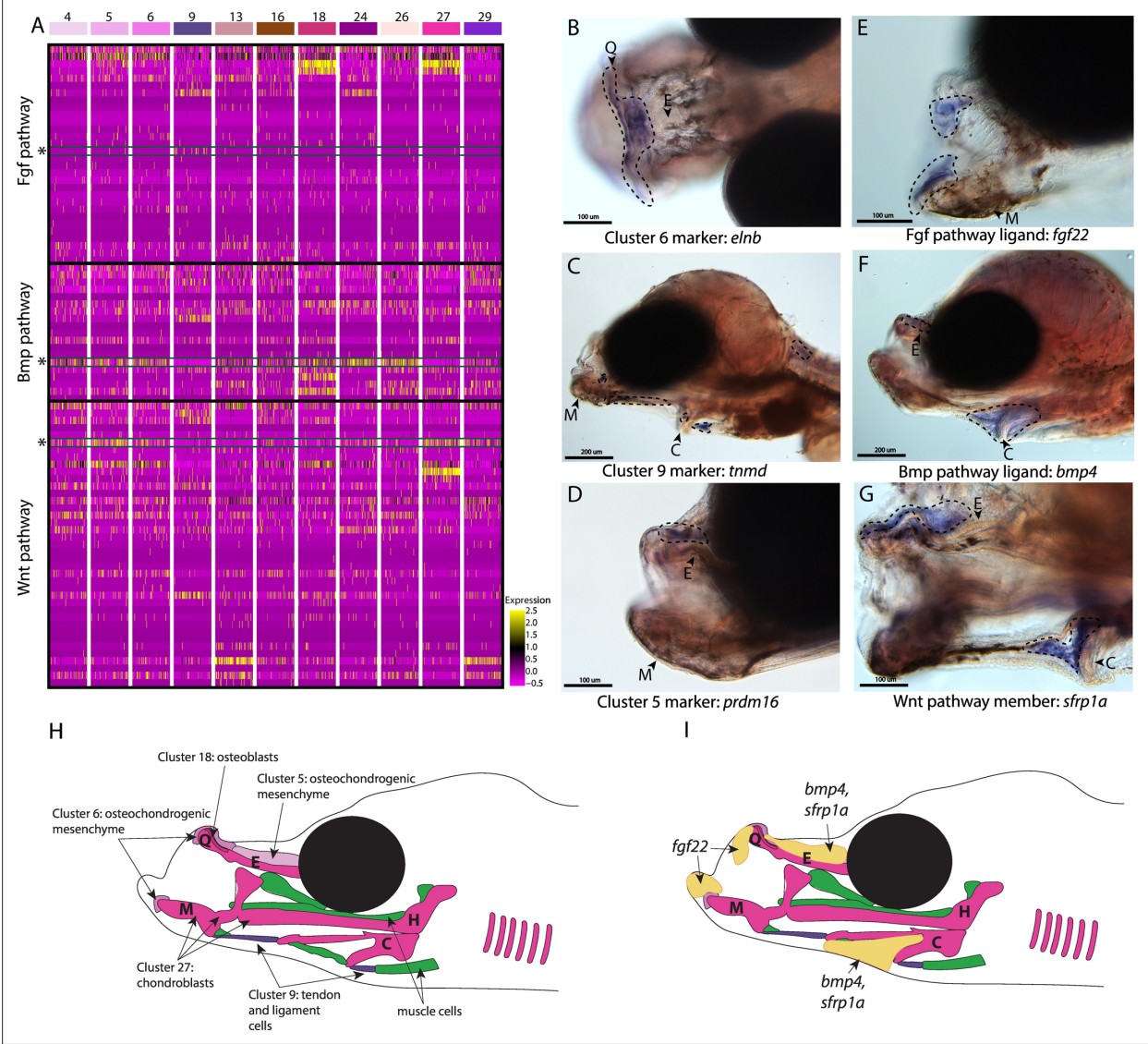

**Figure 4.** Conserved cell types and gene pathways build unique faces of syngnathids. Three main conserved signaling pathways are enriched in connective cell types, MAPK signaling (including Fgf signaling), TGF-beta signaling (including Bmp signaling), and Wnt signaling. Receptors and ligands expression patterns are shown in **A** heatmap from all cell types with cells present in the head. This heatmap features 100 cells downsampled from each cluster and illustrates that many genes from these families are expressed in these cells. Yellow lines indicate high expression of a gene, while hot pink lines indicate no expression. The pathways are boxed in black. Rows representing *fgf22, bmp4,* and *sfrp1a* expression are marked with an asterisk and green box for each respective section of signaling (Fgf, Bmp, and Wnt). Panels **B, C, and D** are *in situ* hybridizations of three marker genes, *elnb, tnmd,* and *prdm16.* The genes *prdm16* and *elnb* mark osteochondrogenic mesenchyme and *tnmd* marks tendons and ligaments. Panels **E, F, and G,** show expression patterns of three pathway representatives *fgf22, bmp4,* and *sfrp1a.* All three genes are expressed in the face: *fgf22* at the tip of the mandible and *bmp4* and *sfrp1a* above the ethmoid and near the ceratohyal. Staining is circled with dashed lines. The Meckel's cartilage (**M**), (mes)Ethmoid cartilage (**E**), Quadrate (**Q**), and Ceratohyal (**C**) are labeled. Panel **C** is a dorsal view. Panels **B, C, E, F, G, H, I** are in lateral view. *In situ* experiments of fgf22 were completed using 10dpf Gulf pipefish. *In situ* experiments of *bmp4, sfrp1a, tnmd, elnb,* and *prdm16* were completed using wild-caught bay pipefish at the onset of craniofacial elongation. Panels **H and I** are summary illustrations of our findings, **H** shows where cells from various clusters were present in the developing head, and **I** illustrates where *bmp4, sfrp1a,* and *fgf22* were expressed. Panel **H** is colored according to cell type.

The online version of this article includes the following figure supplement(s) for figure 4:

**Figure supplement 1.** Conserved fin *prdm16* expression domains in stickleback and pipefish, but fishes have an expression in different regions of the head.

**Figure supplement 2.** Conserved expression domains of *fgf22* in pipefish and stickleback fish.

**Figure supplement 3.** Broad *bmp4* craniofacial expression in stickleback, but *bmp4* is restricted to ethmoid and ceratohyal regions in pipefish.

**Figure supplement 4.** *sfrp1a* has conserved fin expression in pipefish and stickleback, but craniofacial expression differs between the species.

relevant in syngnathid fishes where tendons are enlarged and store elastic energy necessary for their specialized feeding (*Van Wassenbergh et al., 2008*).

The last category of markers contained genes with regulatory roles important for craniofacial development such as *prdm16,* the marker for osteochondro-mesenchymal cells (cluster #5). The gene *prdm16* mediates the methylation of histones and regulates gene expression, key for promoting craniofacial chondrocyte differentiation (*Ding et al., 2013*; *Kaneda-Nakashima et al., 2022*; *Shull et al., 2020*). We found *prdm16* expressed in mesenchymal cells directly above the ethmoid plate and in fins (*Figure 4D*; *Figure 2—figure supplement 63*). Comparison of pipefish with a related, short-snouted fish (stickleback) identified similarity in fin expression, but differences in craniofacial staining (*Figure 4—figure supplement 1*). Stickleback expressed *prdm16* in the hindbrain, gill arches, and lower jaw, consistent with published zebrafish data (*Ding et al., 2013*), but staining was additionally detected in the ethmoid region in pipefish.

We next examined signaling pathways active in craniofacial development. Our KEGG pathway analysis revealed that MAPK, Wnt, and TGF-beta signaling pathways were significantly enriched in one or more craniofacial contributing cell clusters (module #6; *Figures 2B and 3C*, *Figure 4A*). In addition to these KEGG findings, expression of ligands and receptors that are members of these pathways (Fgf, a MAPK pathway, Wnt, and Bmp, a TGF-beta pathway) was observed in cell clusters of both differentiated and undifferentiated states (*Figure 4A*). We chose three genes, one from each major pathway, for *in situ* hybridizations: *fgf22* (a Fgf ligand present in the actively dividing cells' module), *bmp4* (a Bmp ligand present in the largest connective tissue module), and *sfrp1a* (a Wnt pathway enabler present in a cartilage gene module). Although mouse and zebrafish studies identified *fgf22* expression only in the nervous system (*Miyake and Itoh, 2013*; *Umemori et al., 2004*), we found *fgf22* is expressed at the tip of the palatoquadrate and Meckel's cartilage, the gill arches, fins, and brain in pipefish (*Figure 4E*; *Figure 4—figure supplement 2*). These same expression patterns were observed in stickleback, suggesting a surprising co-option of fin and craniofacial expression in percomorph fish.

We observed *bmp4* and *sfrp1a* expressed above the ethmoid plate and along the ceratohyal in pipefish (*Figure 4F–G*). The gene *bmp4* has a conserved role in craniofacial development, particularly important at later stages for driving chondrocyte differentiation (*Wang et al., 2024*; *Zhou et al., 2013*). However, the specificity of *bmp4* expression to mesenchyme around the ceratohyal and ethmoid was not observed in stickleback fish, which had broad craniofacial expression (including the jaws, tooth germs, and gill arches; *Figure 4—figure supplement 3*). Interestingly, *sfrp1a* expression has not been observed in the palate of mice or ethmoid region of zebrafish, but *sfrp1a* craniofacial expression was identified in stickleback (lower jaw and gill arches; *Figure 4—figure supplement 4*) and has been observed in other fishes (*Ahi et al., 2014*; *Schneider et al., 2023a*; *Schilling and Kimmel, 1997*; *Swartz et al., 2011*; *Wang et al., 2024*).

## Gulf pipefish retain tooth development genes but likely lack onset of tooth development

Previous papers have identified possible candidate genes for the loss of teeth in syngnathid fishes including those from genes that initiate tooth bud formation (*fgf4, eve1*), regulate tooth morphogenesis (*fgf3, fgf4*), and synthesize tooth minerals (*scpp4, scpp7, scpp9, odam,* and *scpp5*; *Lin et al., 2016*; *Qu et al., 2021*; *Small et al., 2016*; *Small et al., 2022*; *Zhang et al., 2020*). However, it is unknown whether syngnathid tooth development initiates and then halts or whether it never begins. We searched for signs of early tooth primordia within our atlas to ask whether tooth development might initiate in syngnathids. Additionally, we examined whether genes present in mature teeth are still expressed in syngnathids and what types of cells express them.

Our thorough examination of cell clusters for identity annotation did not find a tooth primordium cluster. We therefore searched for tooth primordia by examining the expression of specific odontogenesis marker genes (*aldh1a2, bmp4, dlx2a, dlx3b, lef1, lhx6a, lhx8, msx1a, msx2,* and *wnt10a*; *Figure 5A*). We observed several primordium genes expressed in our atlas. However, there was no cluster with every marker gene expressed in over 10% of cells. Several markers (*dlx3b, lef1, msx1a, msx2,* and *wnt10a*) were expressed in cluster #28, a fin cluster distinguished by *hoxa13a* and *hoxa13b* expression. Since we previously noted that cluster #16 seems to be a primordial connective tissue cluster, we wondered if it could contain tooth primordial cells. In this cluster, we found the following

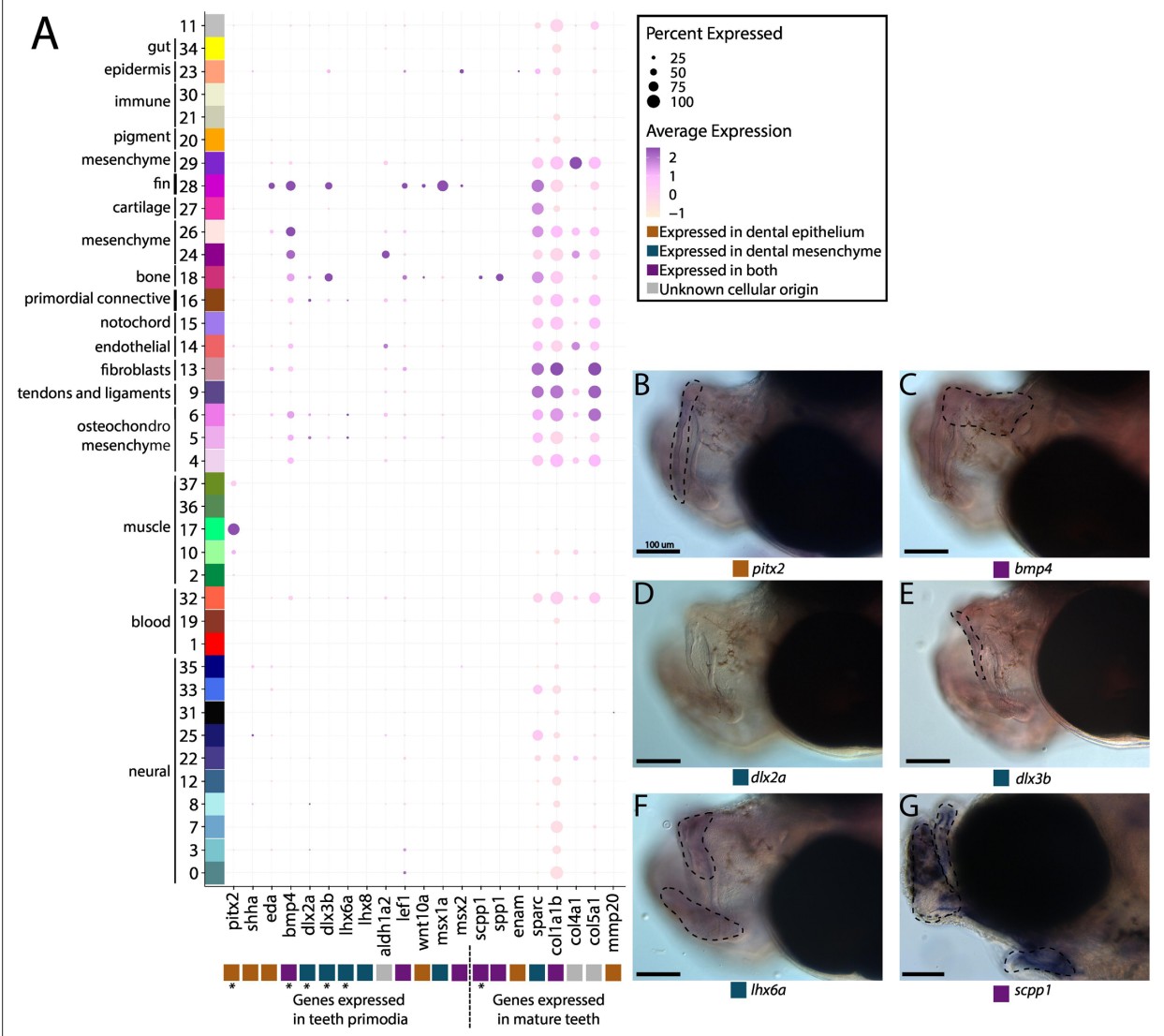

**Figure 5.** Pipefish do not possess identifiable tooth primordium cells, but continue to express tooth development genes in other contexts. Panel **A** presents a dot-plot of genes classified by the tissue layer in which they are reported to be expressed during tooth development in other vertebrates (*Tucker and Sharpe, 2004*; *Gibert et al., 2019*; *Kawasaki, 2009*). The x-axis contains the assayed genes, with asterisks under the genes that were also examined with *in situ* hybridizations. The y-axis contains all the cell clusters. The size of the dot is representative of the percentage of cells from the cluster that express the gene. The color of the dots is an average expression of the gene in the cluster (darker purples represent higher expression). Panel **B** includes *in situ* hybridizations of selected tooth primordia markers (*bmp4, pitx2, lhx6a, dlx2a,* and *dlx3b*) and mature tooth markers (*scpp1*). *In situ* experiments of *bmp4, pitx2, lhx6a, dlx2a,* and *dlx3b* were completed with wild-caught bay pipefish that had begun craniofacial elongation. *In situ* experiments of *scpp1* were completed using 9dpf Gulf pipefish. The scale bars for all images represent 100 µm.

The online version of this article includes the following figure supplement(s) for figure 5:

**Figure supplement 1.** *scpp* gene cluster analysis identifies most *scpp* losses are not unique to syngnathids.

percentage of expression of each gene in the cells: *bmp4* in 46%, *aldh1a2* in 29%, *lef1* in 24%, *dlx2a* in 18%, *dlx3b* in 16.4%, *eda* in 12.7%, *msx1a* in 10.6%, *pitx2* in 10%, *lhx6a* in 8%, *wnt10a* in 4.6%, *msx2* in 2.65%, *lhx8* in 1.8%, and *shha* in 1%.

We suggest that, given the low expression of most tooth marker genes, cluster #16 is unlikely to contain tooth primordial cells. To test this hypothesis, we examined spatial gene expression using *in situ* hybridization of *bmp4, pitx2, lhx6a, dlx2a,* and *dlx3b* in pipefish to ask whether the two definitive cell types are present, namely the dental epithelium (marked by *pitx2* and *bmp4*) and dental mesenchyme (distinguished by *dlx2a, dlx3b,* and *lhx6a*; *Figure 5B–F*; *Gibert et al., 2019*; *Tucker and Sharpe, 2004*). Tooth-specific expression of *dlx2a* is observed solely in the dental mesenchyme

eLife Research article

Developmental Biology | Evolutionary Biology

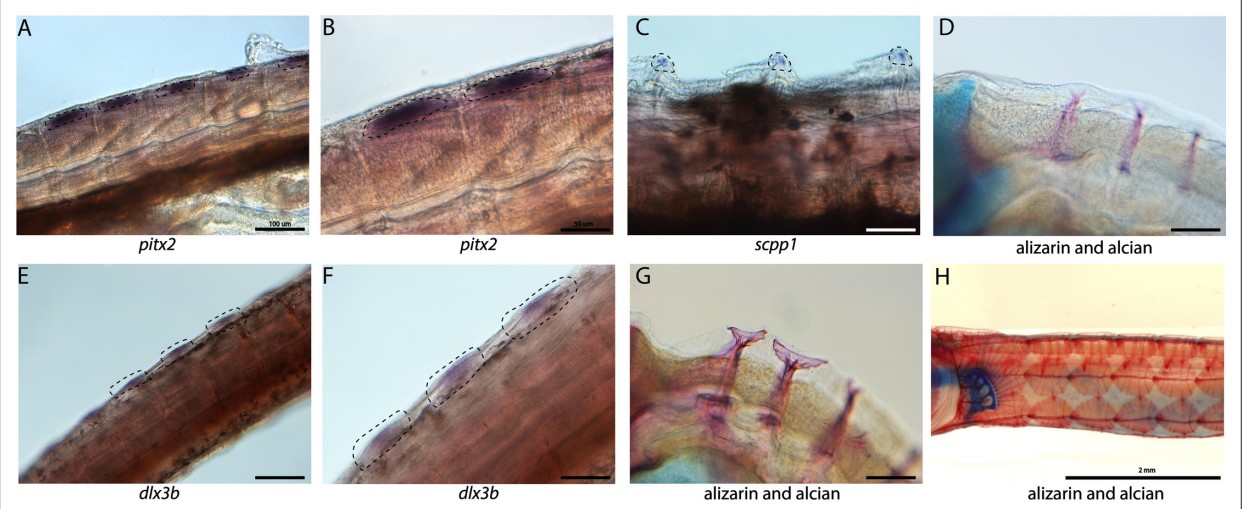

**Figure 6.** Tooth and bone development genes expressed during exoskeleton development. We discovered *pitx2* (**A, B**) and *dlx3b* (**F, E**) expression during the possible emergence of exoskeletal primordium in wild-caught bay pipefish. The embryos used for these *in situ* experiments were in the same stage as those from **Figure 5**, at the beginning of craniofacial elongation. A, E have 100 µm scale bars, B, F have 50 µm scale bars. We further found *scpp1* is expressed at the mineralization front of the exoskeleton in 12dpf Gulf pipefish (**C**) and has a 50 µm scale bar. Alizarin and alcian-stained pipefish are shown in panels **D** (12dpf Gulf pipefish), **G** (1dpf Gulf pipefish), and **H** (adult Gulf pipefish) to illustrate how the exoskeleton forms. D, G have 100 µm scale bars.

in zebrafish and mice, however, it is also expressed in the dental epithelium in medaka (*Stock et al., 2006*). For this study, we labelled *dlx2a* as a mesenchyme marker, though it could be expressed in both dental tissues in a syngnathid outgroup. We found expression of all genes except *dlx2a* in the developing jaws. However, *bmp4*, *pitx2*, *lhx6a*, and *dlx3b* were expressed throughout the face rather than in the punctate pattern observed in tooth primordia development.

We next investigated *scpp* genes (*enam*, *scpp1*, and *spp1*), which are expressed in late tooth development and tooth maintenance in other vertebrates (*Figure 5A*). These genes also have some expression outside of teeth such as in dental bone (*scpp1*, *spp1*; *Kawasaki, 2009*) and fins (*enam*; *Jain et al., 2007*). We identified *spp1* and *scpp1* expression in 54.5% and 24% of bone cells, respectfully, and sparse *spp1* expression (<3%) in other connective tissue cell types. We found minimal expression of *enam* (in 11.4%) in epidermal cells and in less than 8% of muscle cells. In whole mount *in situ* hybridization, we found that *scpp1* is expressed in all developing pipefish bones, both endochondral and dermal (*Figure 5G*). Since *scpp* gene losses observed in syngnathids have been hypothesized to be responsible for their tooth loss (*Lin et al., 2016*; *Qu et al., 2021*; *Zhang et al., 2020*), we explored *scpp* gene family content in the close, tooth-bearing relative to syngnathids, the blue spotted cornetfish, and we found several gene losses (*scpp4*, *scpp7*, and *scpp9*) and a likely pseudogene (*scpp5*) (*Figure 5—figure supplement 1*; *Hughes et al., 2018*; *Stiller et al., 2022*).

Exploration of additional tooth maturation genes (*col1a1b*, *col4a1*, *col5a1*, *sparc*, and *mmp20b*) similarly found that these genes were expressed in non-tooth derivatives, including connective tissue, smooth muscle, and neural cells.

## Tooth and skeletal genes are expressed during dermal armor development

Syngnathid dermal armor is mineralized dermal bone underneath the skin (*Figure 6D, G and H*). It is unknown when syngnathid dermal armor primordia initiate and how they are patterned. Spatial expression analysis of *pitx2* and *dlx3b* in search of tooth primordia instead revealed expression of these genes in possible dermal armor primordia (*Figure 6A, B, E and F*). We found *pitx2* staining localized dorsally to the striated muscle underneath developing dermal armor. The gene *dlx3b* is expressed in a repeating pattern along the body in the epidermal and dermal tissues. Both staining patterns were in discrete regions of the muscle and epidermal layers rather than being continuously

expressed across the tissues. We did not observe the expression of other tooth primordium genes (*bmp4*, *lhx6a*, and *dlx2a*) in this region.

Because the embryos from our atlas had not begun dermal armor mineralization, the atlas cannot be directly used for the discovery of genes active in dermal armor. However, the atlas contains osteoblasts from craniofacial bones which we used to create osteoblast-specific gene networks. We therefore asked whether these osteoblast genes were present in mineralizing dermal armor at later stages. *In situ* hybridization expression analysis revealed that *scpp1*, an osteoblast and tooth mineralization gene, and *ifitm5*, an osteoblast gene, were expressed in the dermal armor at the onset of mineralization (*Figure 6C*; *Figure 2—figure supplement 70*).

## Epithelial expression of immune and nutrient-processing genes may facilitate embryo-paternal interactions in the brood pouch

Within the brood pouch, embryos could interact with male placenta-like tissues, the male brood pouch epithelium, and/or the pouch microbiome. Once the thin chorion is shed, the embryos' epidermis is directly exposed to the pouch environment. We therefore asked if the embryonic epidermal cells expressed nutrient acquisition and/or immune genes that would indicate an active transfer of nutrients and immune response.

Within our larger KEGG analysis, we asked whether nutrient absorption and immune KEGG terms were among the enriched pathways for the epidermal cells. We identified 106 enriched genes in the 'endocytosis' pathway (p-value = 0.036; *Figure 7A*). Four metabolism pathways ('galactose,' 'glutathione,' 'sphingolipid,' and 'starch and sucrose') are also enriched. No immune-related KEGG terms were enriched in the epidermis. For comparison, we investigated whether these KEGG terms are also enriched in the epidermal cells of non-brooding fishes. We completed a KEGG pathway analysis on a comparably staged zebrafish single-cell RNA sequencing atlas (3 d post fertilization; *Lange et al., 2024*). The 3dpf zebrafish epidermal cells did not have a significant enrichment of the 'endocytosis' pathway (23 up-regulated genes, p-value = 0.99) or any metabolism pathway. However, there are 11 endocytosis genes up-regulated in both zebrafish and pipefish epidermal cells, suggesting conserved expression of these genes in pipefish.

We next examined pipefish epidermal gene networks for the presence of nutrient absorption or immune genes. We found that the largest epidermal gene network (#16) contained a striking enrichment of C-type lectin genes, carbohydrate-binding proteins that possess antimicrobial properties (*Figure 7B*). This network contained 14 total lectin genes expressed in the epidermal cells: five galactose-specific lectin nattectin, 2 alpha-N-acetylgalactosamine-specific lectins, one L-rhamnose-binding lectins, four ladder lectin, one C-type lectin 37Dd-like, and one C-type lectin domain family 4 members G-like. Through examining lectin gene expression in the entire dataset, we found that these genes were specific to epidermal cells. Interestingly, previous literature has identified an upregulation of C-type lectins in brood pouch tissues throughout different stages of syngnathid pregnancy (*Roth et al., 2020*; *Small et al., 2013*; *Whittington et al., 2015*).

We did not find any expression of C-type lectin genes in zebrafish epidermal cells (*Figure 7—figure supplement 1*), unlike in pipefish. However, it is possible there are unannotated C-type lectin genes in zebrafish that remained cryptic in the atlas. Interestingly, the 'C-type lectin receptor signaling pathway' was significantly enriched in zebrafish epidermal cells (13 genes, p-value = 0.04) but not in pipefish epidermal cells (34 genes, p-value = 0.4). Although these results seem paradoxical, the KEGG term 'C-type lectin signaling pathway' does not include any of the C-type lectin genes themselves. Overall, these data suggest that the expression of C-type lectin genes in the pipefish embryonic epidermis is potentially unique and warrants further investigation.

## Discussion

Our study examines the development of syngnathids, with a particular focus on novel and adaptive characters, using single-cell RNA sequencing of Gulf pipefish embryos coupled with *in situ* experiments of gene expression. Our single-cell atlas represents early craniofacial skeleton development in Gulf pipefish at a stage when the cartilages of the head skeleton were formed but the face has not elongated. We used the atlas to explore craniofacial and dermal armor development and to investigate potential interactions between the embryos and the brood pouch environment. Our dataset is

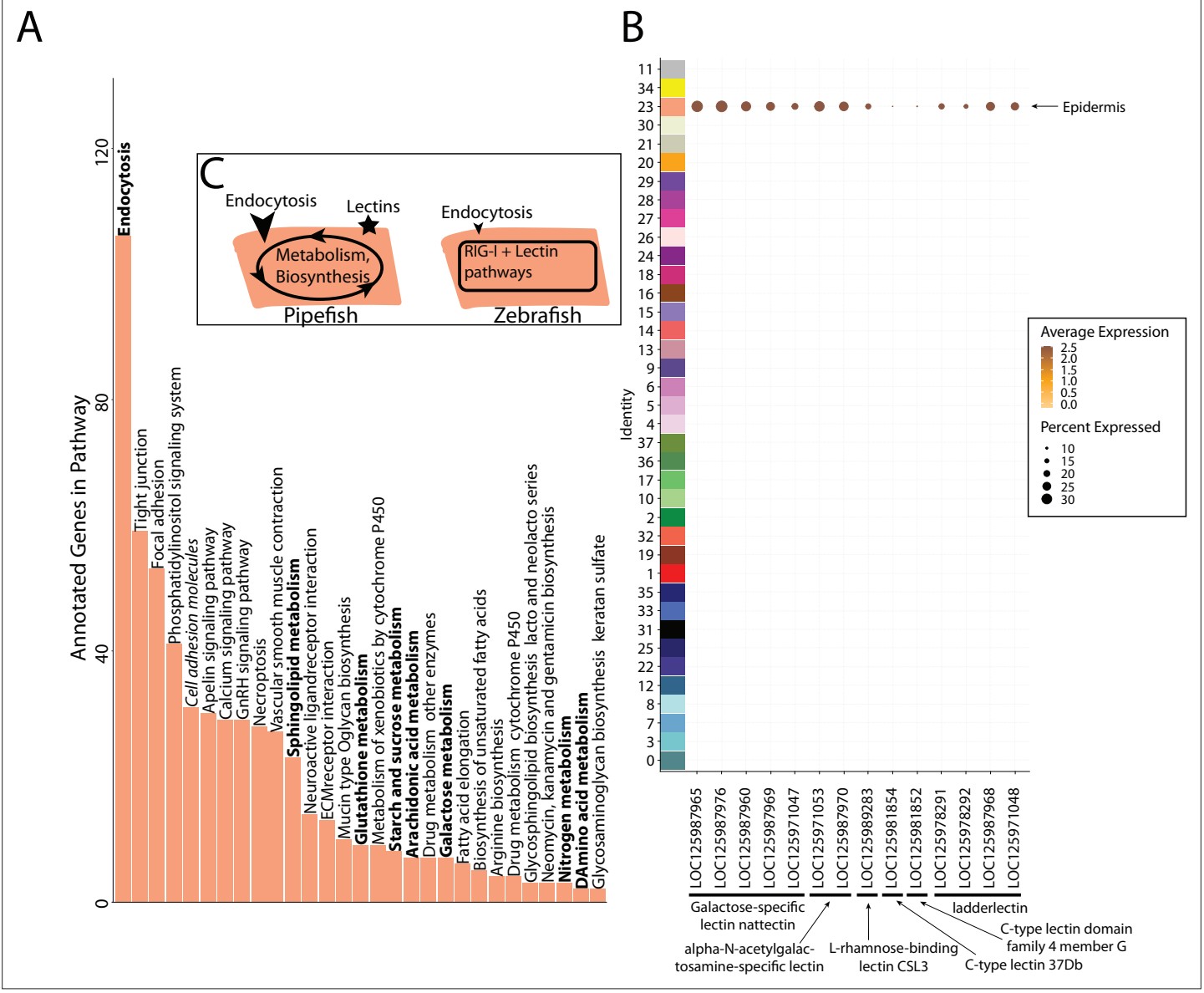

**Figure 7.** Gene expression signatures suggest embryonic interactions within the brood pouch environment. Epidermal cells (**A**), with pathways that suggest increased endocytosis and metabolism pathways are in bold text. Pathways, upregulated in 3dpf zebrafish epidermal cells are in italics. Pipefish epidermal cells also express 15 lectin genes not found in other cell types (**B**). We suggest an epidermal cell model (**C**), in which we predict pipefish have an enrichment of nutrient processing genes and lectins in comparison with zebrafish cells.

The online version of this article includes the following figure supplement(s) for figure 7:

**Figure supplement 1.** 3dpf Zebrafish do not express lectin genes in epidermal cells.

both an opportunity to explore the developmental genetic underpinnings of syngnathid innovations, and a resource for teleost researchers for future studies in this fascinating lineage.

## Our atlas represents a novel resource for Evo-Devo research

Developmental single-cell atlases have elucidated cell identities and genetic pathways active in model organisms such as zebrafish, mice, and chick (*Farnsworth et al., 2020*; *Farrell et al., 2018*; *Feregrino et al., 2019*; *Morrison et al., 2017*; *Soldatov et al., 2019*; *Wagner et al., 2018*; *Williams et al., 2019*). In less traditional models, the majority of scRNAseq atlases are produced from adult tissues, allowing investigations into cell types, population differences, and genetic networks (*Chari et al., 2021*; *Fuess and Bolnick, 2023*; *Hain et al., 2022*; *Hong et al., 2023*; *Koiwai et al., 2023*; *Parker*

*et al., 2022*; *Potts et al., 2022*; *Royan et al., 2021*; *Songco-Casey et al., 2022*; *Vonk et al., 2023*; *Woych et al., 2022*). Creation of developmental scRNAseq atlases in non-model organisms is just beginning to accelerate, but many emerging models still lack such a resource, limiting our understanding of their development (*Healey et al., 2022*; *Salamanca-Díaz et al., 2022*; *Steger et al., 2022*; *Ton et al., 2023*). Our single-cell atlas is one of the first created to understand the development of derived traits in a non-model organism.

For syngnathids specifically, this atlas represents an important step towards understanding the genetic nature of unique syngnathid traits. Numerous developmental genetic changes can lead to evolutionary innovations, including the evolution of novel genes, gene duplications, gene losses, gene family expansions or contractions, evolution of regulatory elements, co-option of gene regulatory networks, re-wiring of gene networks, assembly of novel gene networks, and/or the emergence of novel cells (*Arendt et al., 2016*; *Cañestro et al., 2007*; *Teichmann and Babu, 2004*; *Wagner, 2011*; *Wagner and Lynch, 2010*). Considering these possibilities, we examined select syngnathid traits and speculated on developmental genetic mechanisms influencing their evolution. Continuing to investigate these proposed mechanisms through expanded scRNAseq atlases and other studies will be critical for understanding syngnathid evolution.

## Conserved pathways may contribute to derived syngnathid heads

Syngnathids have highly derived heads including an elongated ethmoid region, uniquely shaped hyoid, and altered muscles and tendons to support specialized 'pivot feeding.' The developmental underpinnings of these derived traits have remained underexplored. In our atlas, we identified cell types present in the developing pipefish head and genetic pathways active in those cell types. We identified numerous cells that were present in the developing face: cartilage, bone, tendons, ligaments, osteochondrogenic mesenchyme, fibroblasts, and unclassified connective tissue cells. Overall, we did not find any unrecognizable cell types, suggesting that genetic modifications within conserved cell types may drive craniofacial modifications.

We next investigated signaling pathways expressed in these cells to determine whether and to what extent developmental genetic reorganization might have occurred. Specifically, we examined the expression of one gene each from three different highly conserved signaling pathways: Wnt (*sfrp1a*), TGF-beta (*bmp4*), and MAPK (*fgf22*). Using *in situ* hybridizations, we found *sfrp1a* and *bmp4* expressed dorsal of the elongating ethmoid plate and surrounding the ceratohyal, suggesting that Wnt and Bmp signaling may be active in the lengthening structures. These two genes are proposed to influence the development of elongated and broadened craniofacial morphologies in other species (*Ahi et al., 2014*; *Schneider et al., 2023a*; *Tucker et al., 2000*). Pipefish *prdm16* is similarly expressed dorsal to the elongating ethmoid plate. Since *prdm16* regulates Wnt and TGF-beta signaling and these genes regulate chondrocyte differentiation (*Bjork et al., 2010*; *Kaneda-Nakashima et al., 2022*; *Shull et al., 2022*; *Wang et al., 2024*), their co-expression might suggest a prolonged period of chondrocyte differentiation along the pipefish ethmoid region. Future studies investigating this testable hypothesis would clarify whether prolonged chondrocyte differentiation broadly underlies craniofacial diversity.

We found *fgf22* expressed in the mandible, gill arches, and fins using *in situ* hybridizations, but not in the elongating regions of the head. Interestingly, *fgf22* expression has not been reported in craniofacial development of any other species (*Miyake and Itoh, 2013*), but we found similar craniofacial and fin expression in stickleback fish. Fgf signaling, however, is a conserved and essential pathway for craniofacial development (*Crump et al., 2004*; *Leerberg et al., 2019*; *McCarthy et al., 2016*; *Szabo-Rogers et al., 2008*; *Walshe and Mason, 2003*; *Woronowicz and Schneider, 2019*), raising the possibility that *fgf22* has been co-opted into a role played by different *fgf* genes in other species. Future work should consider a relationship between the novel craniofacial expression of *fgf22* and the loss of Fgf ligands *fgf3* and *fgf4* in syngnathids. If *fgf22* is active in existing gene networks, particularly those where *fgf3* or *fgf4* is active in other species, then its novel expression may indicate evolved genetic redundancy prior to the gene losses.

Our analysis suggests ways in which unique syngnathid craniofacial structures could have evolved through genetic network evolution. Unusual expression location (e.g. *fgf22*) and specificity (e.g. *sfrp1a*, *bmp4*, and *prdm16*) in pipefish, compared to zebrafish and stickleback, suggests that changes in signaling gene deployment and/or content within craniofacial gene networks,

particularly genes from Wnt, Fgf, or TGF-beta families, could underly the exceptionally elongated syngnathid face.

## Early, not late, tooth development is likely at the root of evolutionary tooth loss

Tooth loss has occurred independently in numerous lineages and has often been studied to understand the developmental basis of character loss. For instance, research in birds and turtles found that tooth programs initiate but are subsequently truncated, explaining toothlessness in mature animals (*Chen et al., 2000*; *Tokita et al., 2013*). Additional studies in birds have found losses in tooth maturation genes (specifically *scpp* genes; *Sire et al., 2008*). Since numerous primordium and maturation genes are lost in syngnathids (*Lin et al., 2016*; *Qu et al., 2021*; *Small et al., 2016*; *Small et al., 2022*; *Zhang et al., 2020*), we asked if syngnathids begin tooth development at all.

We found that early tooth development genes were still expressed in pipefish, which is unsurprising given their pleiotropic roles, but found no convincing evidence either in the atlas or in spatial gene expression analysis of tissues for tooth primordia. The possibility remains, however, that a transient cell population could exist at a different developmental time than assayed here. Barring this caveat, syngnathid *fgf3* and *fgf4* losses could have resulted in insufficient Fgf signaling from the oral epithelium to the dental mesenchyme, causing failure of tooth initiation (*Small et al., 2022*; *Stock et al., 2006*).

If our finding of the loss of the earliest stages of tooth development is consistent across developmental stages and syngnathids, why then have syngnathids retained some members of the *scpp* tooth maturation gene cluster? Since studies in birds propose *scpp* gene losses can occur from relaxed selection (*Sire et al., 2008*), we speculated syngnathids lost *scpp* genes with expression limited to teeth and retained genes with ancestrally pleiotropic expression patterns. In Gulf pipefish, we found the retained genes *scpp1*, *spp1*, and *enam,* are expressed in structures outside of tooth development, suggesting developmental pleiotropic constraint. Specifically, we found *spp1* and *scpp1* expressed in osteoblasts which is consistent with zebrafish (*Bergen et al., 2022*; *Kawasaki, 2009*; *Liu et al., 2016*) and *enam* expressed in the epidermis which has not been reported in zebrafish (*Goldsmith et al., 2003*; *Jain et al., 2007*; *Liu et al., 2016*). Through examining the conservation of the *scpp* genes in close syngnathid relatives, we found that most *scpp* gene losses (*scpp4*, *scpp7*, and *scpp9* and a functional *scpp5*) are shared with a tooth-bearing outgroup to the family, and likely occurred prior to the loss of teeth in syngnathids. Overall, our analysis favors the hypothesis that pleiotropic *scpp* genes were retained in syngnathids while other, more tooth-specific *scpp* genes were lost due to relaxed selection.

## Redeployment of the bone gene network to build dermal armor

The syngnathid dermal armor is a type of evolutionary novelty, which can arise through either differentiation of serially repeated elements or *de novo* origination, derived from either the redeployment of existing gene networks, rewiring of existing gene networks, or the assemblage of new gene networks (*Wagner and Lynch, 2010*). Currently, there is no understanding of the developmental genetic underpinnings of the syngnathid dermal armor.

We identified *dlx3b* and *pitx2* expression in tissues where the dermal plates mineralize later in development. Using *in situ* hybridizations, we showed that epithelial and dermal layers expressed *dlx3b* and underlying muscle cells expressed *pitx2*. In some species, dermal bone, plate, or denticle development occurs from the re-deployment of tooth gene regulatory networks (*Mori and Nakamura, 2022*). However, this does not appear to be the case in syngnathids because the dermal armor lacks the characteristic epithelial–mesenchyme interactions distinguished by *pitx2* expression in the epithelia.

Instead, dermal armor might originate from the co-option of existing bone development gene regulatory networks. Expression of the gene *dlx3b* has been observed in epithelia and mesenchyme during dermal and perichondral bone development in zebrafish (*Verreijdt et al., 2006*). In addition to *dlx3b*, we observed bone development genes *scpp1* and *ifitm5* expressed in the ossifying dermal armor. Future studies could test our hypothesis that the dermal armor evolved through re-deployed osteoblast networks by examining osteoblast gene network expression over time.

## Signatures of embryonic interactions within the novel pouch environment

Syngnathid embryos are reared within the brood pouch, a novel structure and environment composed of male-derived tissues (epithelium and placental-like tissues that include specialized cell types) that harbors a pouch microbiome (*Stölting and Wilson, 2007*). During pregnancy, the male brood pouch undergoes numerous changes including increased vascularization and altered expression of immune genes (*Harada et al., 2022*; *Ripley et al., 2010*; *Roth et al., 2020*; *Small et al., 2013*; *Whittington et al., 2015*). Researchers predict that these changes relate to nutrient and waste transfer and prevention of embryonic rejection and bacterial infection (*Dudley et al., 2021*; *Whittington and Friesen, 2020*). However, there are few studies that examine whether and how embryos interact with the brood pouch environment (*Kvarnemo et al., 2011*; *Ripley and Foran, 2006*). To consider whether embryos have specializations for life in this brood pouch environment, we asked about cell type-specific expression of nutrient acquisition and/or immune genes.

Pipefish embryos uptake paternally derived carbohydrates, proteins, and lipids (*Kvarnemo et al., 2011*; *Ripley and Foran, 2006*). Our data suggest that this uptake could occur through the embryonic epidermis. Specifically, we noticed an enrichment of endocytosis and metabolism genes in epidermal cells. Epidermal absorption of maternally derived nutrients has been suggested in viviparous fishes (*Tengfei et al., 2021*; *Wourms, 1981*). Interestingly, microvilli, a type of cellular projection, have been observed on the anal fin of developing seahorses (*Wetzel and Wourms, 2004*). In light of our findings in pipefish, possibly these seahorse microvilli could be functionally equivalent to those in the small intestine, maximizing nutrient absorption from the environment.

During pregnancy, the male brood pouch increases the expression of C-type lectin genes (*Roth et al., 2020*; *Small et al., 2013*; *Whittington et al., 2015*). These genes are transmembrane or secreted receptors that sense self or non-self and are primarily studied for their role in innate and adaptive immunity (*Brown et al., 2018*). We identified 14 C-type lectin genes expressed in the embryonic epidermis. Our work suggests that lectin genes are produced by both the father and the embryos, but their function is still unclear. Syngnathid research has primarily suggested that lectin genes are produced to prevent bacterial infection (*Melamed et al., 2005*), though they could be important for male-embryo recognition.

Overall, our findings suggest that pipefish embryos have evolved to be specialized for development within the brood pouch by expressing genes related to nutrient acquisition and immunity. Future studies could provide insights into when nutrient acquisition and lectin genes are expressed in development, their functional role, and how their expression varies across syngnathid lineages that have exposed versus enclosed embryos, for example, to examine how embryonic development has been impacted by the brood pouch.

## Conclusions

Our study represents the first scRNAseq developmental atlas in syngnathids, and one of the first non-model developmental scRNAseq atlases, providing a major step forward for evo-devo research. We used our atlas to begin addressing questions on the evolution and development of syngnathid innovations including their unique craniofacial structure, loss of teeth, dermal armor, and development within the male brood pouch. By combining scRNAseq analysis with spatial expression data from *in situ* hybridization, we made important discoveries in cell type identity and distribution as well as spatial expression of marker and signaling genes. We found that syngnathids express genes from conserved signaling pathways during craniofacial development, suggesting that alterations within these pathways may be important for the evolution of their craniofacial skeletons. We did not find evidence for tooth primordia within syngnathids and propose that genetic changes early in tooth development could have led to their loss of teeth. We propose that the re-deployment of bone gene networks, but probably not tooth gene networks, could play a role in the dermal armor development. Finally, we observed an enrichment of endocytosis genes and many C-type lectin genes in epidermal cells, which suggests ways these cells might interact with the brood pouch environment. Our atlas advances our understanding of syngnathid development and evolution and provides resources for developmental genetic analysis in nascent evo-devo model species.

## Methods

### Single-cell RNA sequencing libraries preparation

We created scRNAseq atlases from embryos of wild-caught Gulf pipefish (*Syngnathus scovelli,* acquired from collaborator Emily Rose using Florida Fish and Wildlife collection permit SAL-21–0182-E), and all work was performed according to the University of Oregon approved IACUC protocol (AUP-20–23). Details on the fish, reagents, kits, and primer sequences are provided in the Key Resources table (Appendix 1). We harvested 20 embryos per pouch from two wild-caught male pipefish. Embryos from the same pouch were pooled together to provide two biological replicates. The embryos were at a stage before the tubular face was fully elongated, and while the head skeleton was cartilaginous with minimal signs of mineralization of superficial intramembranous bones. This corresponds to a stage termed 'frontal jaws' in a recent description of pipefish development (*Sommer et al., 2012*).

We dissociated the embryos using 460 ul of 0.25% trypsin in water and 40 ul 100 mg/mL collagenase I (Sigma C0130-200mg) for 16 min. We filtered cells using a 40 uM cell strainer (Thomas Scientific #1181X52). We quantified cell concentrations using the TC20 Automated Cell Counter (Biorad) and then diluted the samples to 800 cells/ul in.04% BSA in PBS. The University of Oregon Genomics and Cell Characterization Core (GC3F; https://gc3f.uoregon.edu) prepared single-cell libraries for each sample using 10X Genomics Single-Cell 3' Genome Expression mRNAseq kit with NextGEM v3.1 chemistry. We sequenced these libraries on an S4 lane on the NovaSeq 6000 at the GC3F. To improve the 3' UTR genome annotations, we also prepared scISOrSeq libraries from the first embryonic sample and from dissociated pouch cells from pregnant and nonpregnant males. These libraries were produced in accordance with (*Healey et al., 2022*). Embryonic, pregnant pouch, and non-pregnant pouch libraries were sequenced separately on PacBio Sequel II - SMRT Cells 8 M.

To turn the scISOrSeq reads into gene models, we followed the pipeline from *Healey et al., 2022*. We ran the script (scISOr_Seq_processing.py from https://github.com/hopehealey/scISOseq_processing; *Healey, 2022*) to remove barcodes, identify cell barcodes, and demultiplex with the single-cell flag and appropriate barcodes (5' CCCATGTACTCTGCGTTGATACCACTGCT and 3' CTACACGACGCTCTTCCGATCT). We aligned the reads to the 2022 Gulf pipefish genome (GenBank: GCA_024217435.2) using minimap v2.9 (*Li, 2018*). We filtered the reads using cDNA cupcake to remove duplicate transcripts (*Tseng, 2021*). We used SQANTI3 to identify gene models and filter them (*Tardaguila et al., 2018*). We merged the SQANTI3 annotations with the Gulf pipefish genome (NCBI GenBank: GCF_024217435.2) using TAMA merge (*Kuo et al., 2017*). Since the Gulf pipefish genome does not contain mitochondrial genes, we appended the annotation and fasta files with the Gulf pipefish mitochondrial genome (NCBI RefSeq: NC_065499.1).

### Single-cell atlas construction

We ran Cell Ranger (10 X Genomics v3.0.2) using our scRNAseq reads, the Gulf pipefish genome assembly with the mitochondrial genome, and the modified gene annotations. Cell Ranger estimated 20,733 cells for sample one, 23,682 genes expressed, and 21,039 mean reads per cell. For sample two, Cell Ranger predicted 17,626 cells, 23,740 genes expressed, and 29,804 mean reads per cell. We analyzed Cell Ranger's output using Seurat (v4.1.0) on R (v4.0.2; *Butler et al., 2018*; *Hafemeister and Satija, 2019*).

To remove extraneous RNA counts from the dataset, we used SoupX (v1.5.2; *Young and Behjati, 2020*). We identified doublet scores for our dataset using scrublet (v0.2.3). The doublet removal step reduced the first sample by 114 cells (from 20,733 cells to 20,619 cells) and the second sample by 167 cells (from 17,626 cells to 17,459 cells). We finally removed cells with less than 500 features, greater than 9000 features, greater than 1E5 RNA counts, with a scrublet score greater than the detected threshold (0.76 for sample 2 and.21 for sample 2), or greater than 10% mitochondrial reads. The second filtering step removed 727 cells from sample one (20,619 cells to 19,892 cells) and 1566 cells from sample two (from 17,459 cells to 15,893 cells).

We normalized the datasets with SCTransform (v0.3.3). We used Seurat's integration tools, SelectIntegrationFeatures using 3000 feature genes, FindIntegrationAnchors using SCT normalization, and IntegrateData using SCT normalization, to integrate the two datasets. After integration, our combined atlas had 35,785 cells (*Supplementary file 1*; *Supplementary file 2*). We then used the integrated dataset to complete the PCA analysis. We tested using a variety of principle components for further

analysis and chose 30 PCs for our analysis based on the clear delineation of major cell types. We next clustered the cells using 30 PCs and plotted the data on a UMAP with Seurat.

## Single-cell atlas cluster identification

To identify cluster identities, we used the RNA assay of the scRNAseq data to find cluster markers with Seurat's FindAllMarkers command with the parameters only.pos=TRUE and logfc.threshold=0.25, requiring markers to be upregulated in the cluster and have a log fold change of at least 0.25. We found a second set of cluster markers through our custom function which searched through all genes and identified genes uniquely expressed in greater than 60% of cells in the cluster and in less than 10% of cells in every other cluster using Seurat's DotPlots. We searched for our identified markers in available zebrafish datasets (*Fabian et al., 2022*; *Farnsworth et al., 2020*; *Lange et al., 2024*), ZFIN (*Howe et al., 2013*), NCBI, https://medlineplus.gov/, and genecards to give the clusters initial annotations. For each cluster, we examined multiple genes using DotPlots and FeaturePlots to propose the cluster identity.

Next, we identified one gene for each cluster which marked the cluster best (expressed in the most cells in the focal cluster and expressed in as few of the other clusters as possible) by consulting the two marker gene lists and examining markers with Dot Plots (*Supplementary file 3*; *Supplementary file 4*; *Supplementary file 5*). Using these markers, we completed a set of *in situ* hybridizations to hone our cluster annotations. Due to challenges in culturing Gulf pipefish, we used both embryos and larvae from *Syngnathus leptorhynchus*, a pipefish from the same genus that lives in Oregon coastal habitats, and allowed for easy collection, and from cultured Gulf pipefish for the cluster annotation *in situ* experiments. We caught a pregnant male *Syngnathus leptorhynchus* using a beach seine near Coos Bay, Oregon under Oregon Department of Fisheries and Wildlife permit number 26987.

*Syngnathus scovelli* used for *in situ* experiments were purchased from Alyssa's Seahorse Saavy and Gulf Specimens Marine Lab and then reared in our facility at 25 ° C water and 25–28 PPT Salinity. We designed probes using NCBI Primer Blast with the Gulf pipefish genome assembly and produced these probes using Gulf pipefish embryonic cDNA pools. To create the probes, we completed two rounds of PCR. The first round used a gene-specific forward primer with a reverse gene-specific primer that had 10 nucleotides of the T7 promotor sequence attached. PCR products were cleaned with Zymo clean and concentrator DNA kit and eluted in 15 µl of elution buffer. Round two of PCR used the same gene-specific forward primer with a modified T7 promoter sequence (TGGACTAATACGACTCACTATAGGG) as the reverse primer, and finally, the product was cleaned again with Zymo clean and concentrator DNA kit and eluted in 15 ul of elution buffer.

Round one PCR conditions are as follows: 95 degrees Celsius for 3:00 min, 40 cycles of denaturation (95 degrees for 30 s), annealing (30 s, annealing temperature varied by probe), and extension (72 degrees for 1:00 min), and a final extension step of 3:00 min. Round two PCR conditions are as follows: 98 degrees Celsius for 30 s, 30 degrees for 10 s, 72 degrees for 50 ss, then 35 rounds of denaturation (98 degrees for 10 s), annealing (50 degrees for 10 s), and extension (72 degrees for 50 s), and a final extension step of 10 min at 72 degrees. For round two PCRs with multiple bands (specifically, *ifitm5*), the band of the expected size was excised with a razorblade and DNA was extracted with a Zymoclean gel DNA recovery kit.

The round two PCR product was sanger sequenced to confirm identity. All the marker gene primers and successful PCR conditions are in *Supplementary file 6*. Alignments of the probes to the unpublished *Syngnathus leptorhynchus* genome assembly are included in *Supplementary file 7*. The probes were transcribed with T7 polymerase for 2–6 hr then cleaned with Zymo RNA clean and concentrator and eluted into 30 µl of water. For the *in situ* hybridizations, we selected embryos and newly spawned larvae close to the developmental stage used in the atlas. We completed *in situ* hybridizations in keeping with (*Thisse and Thisse, 2008*), leaving the embryos in stain until the background was observed. For spawned larvae, we completed a bleaching step (1% H2O2 and 0.5% KOH for 8 min) prior to the proteinase K digest. After imaging, we used the levels tool in Adobe Photoshop (v23.4.2) to white-balance the pictures.

## Single-cell KEGG analysis

To identify pathways upregulated in cell clusters, we completed a KEGG analysis. We downloaded Gulf pipefish KEGG pathways (https://www.kegg.jp). For the KEGG analysis, we used the marker

genes identified from our FindAllMarkers list as the input genes. We converted these gene ids using keggConv to KEGG ids. We used a Wilcoxon enrichment test to ask whether cluster marker genes were enriched for each KEGG pathway.

## Differentiation state analysis

To assess whether proposed primordial cell clusters were composed of undifferentiated cells relative to other clusters from their lineages, we completed a differentiation analysis using CytoTRACE (v0.3.3, *Gulati et al., 2020*). Clusters from similar lineages (neural: 0, 3, 7, 8, 12, 22, 25, 33, and 35; muscle: 2, 10, 17, and 37; connective: 4, 5, 6, 9, 15, 16, 18, 24, 27, 28, and 29) were isolated using subset. Cell counts were gathered using as.Matrix(GetAssayData), then CytoTRACE was run on these counts. Data was plotted on a VlnPlot to show the variability.

## Single-cell atlas network analysis

To identify genetic networks present in our atlas, we completed a weighted gene network correlation analysis using WGCNA (v1.72–1, *Langfelder and Horvath, 2008*). We selected 3000 variable features from the integrated assay of the single-cell dataset for the WGCNA. We created an adjacency matrix from the data using bicor with a maxPOutliers of 0.05. To decide on a beta value or the soft threshold power, we created an adjacency matrix plot using pickSoftThreshold and picked the threshold where the scale-free topology model fit leveled off. We selected the value of two and we then raised the adjacency matrix to the power of two.

We calculated the dissimilarity matrix by calculating the TOM similarity of the adjacency matrix and subtracting it from one. We then created a gene tree through hclust of the dissimilarity matrix with the average method. From the gene tree, we created modules using cutreeDynamic with a deep Split of two and the minClusterSize of 15 (setting the smallest cluster size to 15). We calculated module eigengenes using moduleEigengenes of the adjacency matrix then calculated the dissimilarity of the module eigengenes with cor of the module eigengenes subtracted from one and clustered the module eigengenes with hclust. We then chose a dissimilarity of 35% as the cut-off for merging modules; however, no modules had dissimilarity scores with each other below 40%. Since the module eigengenes are calculated for each cell in the dataset, we used these values to calculate the t-statistic for each cell cluster (considered each sample) in the module (all of the module eigengene scores were used as the population).

We next tested the hypothesis that certain cell clusters were strongly associated with specific gene modules through a two-way permutation test (1000 permutations) and corrected p-values to control the false discovery rate (FDR). Additionally, we tested whether specific cell clusters drove underlying gene network structure by measuring network connectivity. The network connectivity was first calculated using the entire dataset, then each cell cluster was progressively dropped from the dataset and the connectivity was remeasured. This resulted in change in a connectivity scores for each module-cluster pair. To assess whether these changes were significant, we completed 1000 permutations whereby cells were randomly dropped (the number of cells dropped was equal to the cell cluster size of the focal cluster), connectivity was measured, and the change in connectivity was recorded. The p-value is the number of instances where the change in connectivity is greater in the permutations than in the focal cell cluster run. P-values were corrected to control the FDR.

Using the module gene lists, we identified the number of genes from each module that were found in each KEGG pathway. Since the KEGG modules do not have p-values associated with the genes, we could not complete a Wilcoxon enrichment test. We instead removed any pathways where there were less than three genes present for any pathway and noted that there was no statistical test run on these KEGG results. We visualized the networks using Cytoscape (v3.10.0).

## Zebrafish data analysis

We downloaded a 3dpf zebrafish scRNAseq atlas from *Lange et al., 2024*. Marker genes were identified using FindAllMarkers. In accordance with the pipefish KEGG analysis, we detected enriched KEGG pathways using zebrafish KEGG terms and these marker genes. Zebrafish lectin genes were identified on NCBI, and their expression was visualized via DotPlots.

## *In situ* hybridization

For genes chosen for further follow-up analysis, we completed *in situ* hybridizations in Gulf pipefish (*Syngnathus scovelli*) or bay pipefish (*Syngnathus leptorhynchus*). Primer sequences were designed

using NCBI Primer Blast with the Gulf Pipefish genome and synthesized using Gulf pipefish embryonic cDNA. The *sfrp1a* probe was synthesized using the PCR-based probe preparation protocol described in the Single-Cell Identification section. All other probes (*bmp4*, *dlx2a*, *dlx3b*, *fgf22*, *lhx6a*, *pitx2*, and *scpp1*) were prepared using TOPO cloning. Probe primer sequences as well as the species used for the *in situ* experiments are described in *Supplementary file 6*. Pregnant male bay pipefish were caught as described above. These fish were euthanized with MS-222 in accordance with IACUC-approved protocols, then embryos were removed from the brood pouch. For Gulf pipefish embryos, we allowed the fish to mate in our facility and then harvested embryos once they reached the appropriate stages. Fish were reared in 25 °C water with 25–28 ppt salinity.

Select genes with craniofacial expression were additionally probed in threespine stickleback (*Gasterosteus aculeatus*). Primers were designed using NCBI Blast with the threespine stickleback genome and synthesized with stickleback embryonic cDNA. The probes were synthesized using the PCR-based probe preparation protocol described in the Single-Cell Identification section (*Supplementary file 6*). To generate embryos, we crossed a laboratory line of stickleback isolated from Cushman Slough (Oregon) using standard procedures from the Cresko Laboratory Stickleback Facility (*Cresko et al., 2004*). Fish were reared at 20 °C until 9 d post fertilization. Then, fish were euthanized with MS-222 following IACUC-approved procedures.

Embryos and larvae were fixed in 4% PFA, dehydrated through a series of PBT/MeOH washes, and stored in MeOH at –20 C. 5–12 embryos were used for each probe. We completed *in situ* hybridizations in keeping with (*Thisse and Thisse, 2008*), leaving the embryos in stain until the background was observed. After imaging, we used the levels tool in Adobe Photoshop (v23.4.2) to white-balance the pictures.

## Bone and cartilage staining

We used alcian and alizarin stains to mark cartilage and bones. Specifically, we assayed cartilage and bone development in siblings of the scRNAseq samples. These embryos were fixed in 4% PFA and then stored at –20 C in MeOH. We followed the protocol from *Walker and Kimmel, 2007* with minor alterations. We stored samples in 50% glycerol/0.1% KOH at 4 C and imaged them in 100% glycerol. After imaging, we white-balanced the photographs using Photoshop (v23.4.2) levels tool.

## Gene cluster analysis

To examine close syngnathid outgroups, we downloaded the 2023 mandarin dragonet genome (GenBank assembly accession: GCA_027744825.1) and the 2024 cornetfish genome assembly (GenBank assembly accession: GCA_037954325.1) from NCBI. Since these genomes were unannotated, we manually identified *scpp* genes. We searched for *scpp* genes using BLASTN with medaka and additional fish sequences as the query. We also searched for these genes with mVISTA plots across conserved gene synteny regions (LAGAN alignment using translated anchoring) with medaka as the focal species (*Frazer et al., 2004*; *Mayor et al., 2000*). To identify *scpp* genes in additional species, we gathered cluster information from NCBI and ensembl. Additionally, we used mVISTA plots to further search for unannotated *scpp* genes.

## Acknowledgements

We are indebted to Emily Rose for her collection of the Gulf pipefish specimens for the single-cell libraries. We are grateful to everyone in the Cresko Lab for their ideas and participation in early morning pipefish collection trips. We are additionally thankful to Emily Beck for her assistance in collecting pipefish. In particular, we are immensely appreciative of Mark Currey and Tiffany Thornton's valiant efforts to culture the Gulf pipefish. Additionally, we are grateful to Balan Ramesh and Adam Jones for collaborating with the Cresko Lab to produce new Syngnathiformes genomes. We thank Tina Arredondo from the UO GC3F for her preparation of our single-cell RNA sequencing libraries. This work was funded by the National Science Foundation Grant OPP-2015301 (to WAC, SB, and CMS), University of Oregon Research Excellence funds (WAC), and National Institute of Health Fellowship F31DE032559-02 (to HMH). Additionally, HMH was supported by the Genetics Training Program (NIH T32GM149387). Finally, VG received funding and mentorship from the Hui Undergraduate Research Scholars and MW was supported by the Knight Campus Undergraduate Scholars program at the University of Oregon.

# Additional information

## Funding

| Funder | Grant reference number | Author |
| --- | --- | --- |
| National Institute of Dental and Craniofacial Research | F31DE032559-02 | Hope M Healey |
| National Science Foundation | OPP-2015301 | William A Cresko |
| National Institutes of Health | T32GM149387 | Hope M Healey |
| University of Oregon | Hui Undergraduate Research Scholars | Vithika Goyal |
| University of Oregon | the Knight Campus Undergraduate Scholars program | Micah A Woods |
| University of Oregon | Research Excellence funds | William A Cresko |

The funders had no role in study design, data collection and interpretation, or the decision to submit the work for publication.

## Author contributions

Hope M Healey, Conceptualization, Data curation, Software, Formal analysis, Supervision, Funding acquisition, Validation, Investigation, Visualization, Methodology, Writing – original draft, Writing – review and editing; Hayden B Penn, Vithika Goyal, Micah A Woods, Investigation, Writing – review and editing; Clayton M Small, Investigation, Methodology, Writing – review and editing; Susan Bassham, Methodology, Writing – review and editing; William A Cresko, Conceptualization, Resources, Supervision, Funding acquisition, Writing – review and editing

## Author ORCIDs

Hope M Healey ⓘ https://orcid.org/0000-0001-9978-1553
Hayden B Penn ⓘ https://orcid.org/0009-0000-3090-5423
Clayton M Small ⓘ https://orcid.org/0000-0003-1615-7590
Susan Bassham ⓘ https://orcid.org/0000-0002-7309-2095
Vithika Goyal ⓘ https://orcid.org/0009-0009-8601-8474
Micah A Woods ⓘ https://orcid.org/0009-0004-2156-3352
William A Cresko ⓘ https://orcid.org/0000-0002-3496-8074

## Ethics

All of the work was performed in strict accordance to the approved institutional animal care and use committee (IACUC) protocol (AUP-20-23) of the University of Oregon. Pipefish were collected under approved permits (Syngnathus scovelli, Florida Fish and Wildlife permit SAL-21-0182-E; Syngnathus leptorhynchus, Oregon Department of Fisheries and Wildlife permit 26987).

Reviewer #1 (Public review): https://doi.org/10.7554/eLife.97764.3.sa1
Reviewer #2 (Public review): https://doi.org/10.7554/eLife.97764.3.sa2
Reviewer #3 (Public review): https://doi.org/10.7554/eLife.97764.3.sa3
Author response https://doi.org/10.7554/eLife.97764.3.sa4

# Additional files

## Supplementary files

Supplementary file 1. Quality metrics for the single-cell libraries.

Supplementary file 2. The number of cells in each cell cluster and cluster identities.

Supplementary file 4. Marker genes identified using the DotPlot method.

Supplementary file 5. Additional information on the marker gene identified for every cluster.

Supplementary file 6. A list of the *in situ* hybridization probes used in this study, the conditions used to prepare the probes, and the staging/sample information for the embryos.

Supplementary file 7. Alignments of *in situ* hybridization probes with the unpublished *Syngnathus leptorhynchus* genome.

Supplementary file 8. Genetic networks were initially labeled with colors, we converted these labels to numeric annotations for simplicity using this conversion table. The table also contains the number of genes in each network.

Supplementary file 9. All the genetic networks, the genes inside of them, and additional information for the genetic networks highlighted in this paper.

Supplementary file 10. The t-statistics derived for each module-cell cluster pair. The cell clusters are in the rows and the gene modules are in the columns.

Supplementary file 11. p-values for the t-statistics of the strength of association between gene modules and cell clusters. p-values are corrected for multiple testing hypotheses using fdr. The cell clusters are in the rows and the gene modules are in the columns.

Supplementary file 12. The change in connectivity for gene modules when individual cell clusters are removed. The cell clusters are in the columns and the gene networks are in the rows.

MDAR checklist

## Data availability

All raw sequencing data associated with this study are published via NCBI (PRJNA1168967). The integrated single cell RNA sequencing atlas is also available through NCBI (GSE278814). The fasta file and updated Gulf pipefish annotation are stored on Dryad. Code used for the analysis is available on GitHub, copy archived at *Healey, 2024*.

The following datasets were generated:

| Author(s) | Year | Dataset title | Dataset URL | Database and Identifier |
|---|---|---|---|---|
| Healey H, Penn H, Small C, Bassham S, Goyal V, Woods M, Cresko W | 2024 | Single cell RNA sequencing provides clues for the developmental genetic basis of Syngnathidae's evolutionary adaptations | https://doi.org/10.5061/dryad.f7m0cfz60 | Dryad Digital Repository, 10.5061/dryad.f7m0cfz60 |
| Healey H, Penn H, Small C, Bassham S, Goyal V, Woods M, Cresko W | 2024 | Single Cell RNA Sequencing Provides Clues for the Developmental Genetic Basis of Syngnathid Fish Evolutionary Adaptations (Gulf pipefish) | https://www.ncbi.nlm.nih.gov/bioproject/?term=PRJNA1168967 | NCBI BioProject, PRJNA1168967 |
| Healey H, Penn H, Small C, Bassham S, Goyal V, Woods M, Cresko W | 2024 | Single Cell RNA Sequencing Provides Clues for the Developmental Genetic Basis of Syngnathid Fish Evolutionary Adaptations | https://www.ncbi.nlm.nih.gov/geo/query/acc.cgi?acc=GSE278814 | NCBI Gene Expression Omnibus, GSE278814 |

The following previously published dataset was used:

| Author(s) | Year | Dataset title | Dataset URL | Database and Identifier |
|---|---|---|---|---|
| Lange M, Granados A, VijayKumar S, Bragantini J, Ancheta S, Santhosh S, Borja M, Kobayashi H, McGeever E, Solak AC, Yang B, Zhao X, Liu Y, Detweiler AM, Paul S, Mekonen H, Lao T, Banks R, Kim YJ, Royer LA | 2024 | Zebrahub: Multimodal Zebrafish Developmental Atlas | https://www.ncbi.nlm.nih.gov/bioproject/?term=PRJNA940501 | NCBI BioProject, PRJNA940501 |

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

# Appendix 1

## Appendix 1—key resources table

| Reagent type (species) or resource | Designation | Source or reference | Identifiers | Additional information |
|---|---|---|---|---|
| Biological sample (*Syngnathus scovelli*) | *S. scovelli* embryos | Dr. Emily Rose, Alyssa's Seahorse Saavy, and Gulf Specimens Marine Lab | | Florida Fish and Wildlife Collection permit SAL -21–0182-E (Dr. Rose) |
| Biological sample (*Syngnathus leptorhynchus*) | *S. leptorhynchus* embryos | Oregon Coast | | Oregon Department of Fisheries and Wildlife Collection permit 26987 |
| Chemical compound | collagenase I | Sigma | C0130-200mg | 100 mg/ml |
| Commercial assay or kit | DNA Clean and Concentrator | Zymo | D4013 | |
| Commercial assay or kit | RNA Clean and Concentrator | Zymo | R1013 | |
| Commercial assay or kit | Zymoclean gel DNA recovery | Zymo | D4001 | |
| Commercial assay or kit | Next GEM Single-Cell Gene Expression 3' v3.1 (Dual Index) | 10 X Genomics | PN-1000121 | |
| Commercial assay or kit | SMRTbell Express Template Prep Kit 2.0 | PacBio | 100-938-900 | |
| Commercial assay or kit | Barcoded Adapter Kit 8B-OVERHANG | PacBio | 101-628-500 | |
| Other | 10 uM cell strainer | Thomas Scientific | 1181X52 | Cell strainers were used to filter dissociated cells. |
| Software, algorithm | R | R | RRID:SCR_001905 | v4.0.2 |
| Software, algorithm | minimap | *Li, 2018* | | v2.9 |
| Software, algorithm | cDNA cupcake | *Tseng, 2021* | | |
| Software, algorithm | SQANTI3 | *Tardaguila et al., 2018* | | |
| Software, algorithm | TAMA | *Kuo et al., 2017* | | |
| Software, algorithm | Cell Ranger | 10 X Genomics | RRID:SCR_017344 | v3.0.2 |
| Software, algorithm | Seurat | *Butler et al., 2018*; *Hafemeister and Satija, 2019* | | v4.1.0 |
| Software, algorithm | WGCNA | *Langfelder and Horvath, 2008* | | v1.72–1 |
| Software, algorithm | SoupX | *Young and Behjati, 2020* | | v1.5.2 |
| Software, algorithm | scrublet | *Wolock et al., 2019* | | v0.2.3 |
| Software, algorithm | SCTransform | *Hafemeister and Satija, 2019* | | v0.3.3 |
| Software, algorithm | CytoTRACE | *Gulati et al., 2020* | | v0.3.3 |
| Software, algorithm | Cytoscape | | RRID:SCR_003032 | v3.10.0 |
| Software, algorithm | mVISTA | *Frazer et al., 2004*; *Mayor et al., 2000* | | |
| Software, algorithm | Photoshop | Adobe | RRID:SCR_014199 | v23.4.2 |

