## [Editor Report · eLife Assessment]

This study provides a **valuable** new resource to investigate the molecular basis of the particular features characterizing the pipefish embryo. The authors found both unique and shared gene expression patterns in pipefish organs compared with other teleost fishes. The **solid** data collected in this unconventional model organism will give new insights into understanding the extraordinary adaptations of the Syngnathidae family and will be of interest in the domain of evolution of fish development.

---

## [Referee Report · Reviewer #1 (Public review)]

Syngnathid fishes (seahorses, pipefishes, and seadragons) present very particular and elaborated features among teleosts and a major challenge is to understand the cellular and molecular mechanisms that permitted such innovations and adaptations. The study provides a valuable new resource to investigate the morphogenetic basis of four main traits characterizing syngnathids, including the elongated snout, toothlessness, dermal armor and male pregnancy. More particularly, the authors have focused on a late stage of pipefish organogenesis to perform single-cell RNA-sequencing (scRNA-seq) completed by in situ hybridization analyses to identify molecular pathways implicated in the formation of the different specific traits.

The first set of data explores the scRNA-seq atlas composed of 35,785 cells from two samples of gulf pipefish embryos that authors have been able to classify into major cell types characterizing vertebrate organogenesis, including epithelial, connective, neural and muscle progenitors. To affirm identities and discover potential properties of clusters, authors primarily use KEGG analysis that reveals enriched genetic pathways in each cell types. After revisions, the authors have provided extended supplementary files to well interpret the dataset and some statements have been clarified. I thank the authors for the revisions/completions of ISH results compared to initial submission.

To conclude, the scRNA-seq dataset in this unconventional model organism will be useful for the community and will provide clues for future research to understand the extraordinary evolution of the Syngnathidae family.

---

## [Referee Report · Reviewer #2 (Public review)]

Summary:

The authors present the first single-cell atlas for syngathid fishes, providing a resource for future evolution & development studies in this group.

Strengths:

The concept here is simple and I find the manuscript to be well written. I like the in situ hybridization of marker genes >> this is really nice. I also appreciate the gene co-expression analysis to identify modules of expression. There are no explicit hypotheses tested in the manuscript, but the discovery of these cell types should have value in this organism and in the determination of morphological novelties in seahorses and their relatives.

Weaknesses:

I think there are a few computational analyses that might improve the generality of the results.

(1) The cell types: The authors use marker gene analysis and KEGG pathways to identify cell types. I'd suggest a tool like SAMap (https://elifesciences.org/articles/66747) which compares single cell data sets from distinct organisms to identify 'homologous' cell types -- I imagine the zebrafish developmental atlases could serve as a reasonable comparative reference.

(2) Trajectory analyses: Authors suggest that their analyses might identify progenitor cell states and perhaps related differentiated states. They might explore cytoTRACE and/or pseudotime-based trajectory analyses to more fully delineate these ideas.

(3) Cell-cell communication: I think it's very difficult to identify 'tooth primordium' cell types, because cell types won't be defined by organ in this way. for instance dental glia will cluster with other glia, dental mesenchyme will likely cluster with other mesenchymal cell types. so the histology and ISH in most convincing in this regard. having said this, given the known signaling interactions in the developing tooth (and in development generally) the authors might explore cell-cell communication analysis (e.g., CellChat) to identify cell types that may be interacting.

Comments on revisions:

I feel essentially the same about this manuscript. it's a useful resource for future experimental forays into this unique system. The team made improvements to deal with comments from other reviewers related to quality of confirmatory in situ hybridization. This is good.

Regarding their response that one can't use CellChat if you're not working in mice or human, this is inaccurate. the assumption one makes is that ligand-receptor pairs and signaling pathways have conserved functions across animals (vertebrates). It's the same assumption the authors make when using the KEGG pathway to score enrichment of pathways in clusters. CellChat used in fishes in Johnson et al 2023 Nature Communications | (2023) 14:4891.

---

## [Referee Report · Reviewer #3 (Public review)]

Summary:

This study established a single-cell RNA sequencing atlas of pipefish embryos. The results obtained identified unique gene expression patterns for pipefish-specific characteristics, such as fgf22 in the tip of the palatoquadrate and Meckel's cartilage, broadly informing the genetic mechanisms underlying morphological novelty in teleost fishes. The data obtained are unique and novel, potentially important in understanding fish diversity. Thus, I would enthusiastically support this manuscript if the authors improve it to generate stronger and more convincing conclusions than the current forms.

Weakness:

Regarding the expression of sfrp1a and bmp4 dorsal to the elongating ethmoid plate and surrounding the ceratohyal: Are their expression patterns spatially extended or broader compared to the pipefish ancestor? Is there a much closer species available to compare gene expression patterns with pipefish? Did the authors consider using other species closely related to pipefish for ISH? Sfrp1a and bmp4 may be expressed in the same regions of much more closely related species without face elongation. I understand that embryos of such species are not always accessible, but it is also hard to argue responsible genes for a specific phenotype by only comparing gene expression patterns between distantly related species (e.g., pipefish vs. zebrafish). Due to the same reason, I would not directly compare/argue gene expression patterns between pipefish and mice, although I should admit that mice gene expression patterns are sometimes helpful to make a hypothesis of fish evolution. Alternatively, can the authors conduct ISH in other species of pipefish? If the expression patterns of sfrp1a and bmp4 are common among fishes with face elongation, the conclusion would become more solid. If these embryos are not available, is it possible to reduce the amount of Wnt and BMP signal using Crispr/Cas, MO, or chemical inhibitor? I do think that there are several ways to test the Wnt and/or BMP hypothesis in face elongation.

---

## [Author Response]

The following is the authors’ response to the original reviews.

**Public Reviews:**

**Reviewer #1 (Public Review):**
Syngnathid fishes (seahorses, pipefishes, and seadragons) present very particular and elaborated features among teleosts and a major challenge is to understand the cellular and molecular mechanisms that permitted such innovations and adaptations. The study provides a valuable new resource to investigate the morphogenetic basis of four main traits characterizing syngnathids, including the elongated snout, toothlessness, dermal armor, and male pregnancy. More particularly, the authors have focused on a late stage of pipefish organogenesis to perform single-cell RNA-sequencing (scRNA-seq) completed by in situ hybridization analyses to identify molecular pathways implicated in the formation of the different specific traits.The first set of data explores the scRNA-seq atlas composed of 35,785 cells from two samples of gulf pipefish embryos that authors have been able to classify into major cell types characterizing vertebrate organogenesis, including epithelial, connective, neural, and muscle progenitors. To affirm identities and discover potential properties of clusters, authors primarily use KEGG analysis that reveals enriched genetic pathways in each cell types. While the analysis is informative and could be useful for the community, some interpretations appear superficial and data must be completed to confirm identities and properties. Notably, supplementary information should be provided to show quality control data corresponding to the final cell atlas including the UMAP showing the sample source of the cells, violin plots of gene count, UMI count, and mitochondrial fraction for the overall

dataset and by cluster, and expression profiles on UMAP of selected markers characterizing cluster identities.

We thank the reviewer for these suggestions, and have added several figures and supplemental files in response. We added a supplemental UMAP showing the sample that each cell originated (S1). We also added supplemental violin plots for each sample showing the gene count, unique molecular identifier (UMI) count, mitochondrial fraction, and the doublet scores (S2). We added feature plots of zebrafish marker genes for these major cell types and marker genes identified from our dataset to the supplement (S3:S57). We also provided two supplemental files with marker genes. These changes should clarify the work that went into labeling the clusters. Although some of the cluster labels are general, we decided it would be unwise to label clusters with speculated specific annotations. We only gave specific annotations to clusters with concrete markers and/or in situ hybridization (ISH) results that cemented an annotation. As shown in the new supplemental figures and files, certain clusters had clear, specific markers while others did not. Therefore, we used caution when we annotated clusters without distinct markers.

The second set of data aims to correlate the scRNA-seq analysis with in situ hybridizations (ISH) in two different pipefish (gulf and bay) species to identify and characterize markers spatially, and validate cell types and signaling pathways active in them. While the approach is rational, the authors must complete the data and optimize labeling protocols to support their statements. One major concern is the quality of ISH stainings and images; embryos show a high degree of pigmentation that could hide part of the expression profile, and only subparts and hardly detectable tissues/stainings are presented. The authors should provide clear and good-quality images of ISH labeling on whole-mount specimens, highlighting the magnification regions and all other organs/structures (positive controls) expressing the marker of interest along the axis. Moreover, ISH probes have been designed and produced on gulf pipefish genome and cDNA respectively, while ISH labeling has been performed indifferently on bay or gulf pipefish embryos and larvae. The authors should specify stages and species on figure panels and should ensure sequence alignment of the probe-targeted sequences in the two species to validate ISH stainings in the bay pipefish. Moreover, spatiotemporal gene expression being a very dynamic process during embryogenesis, interpretations based on undefined embryonic and larval stages of pipefish development and compared to 3dpf zebrafish are insufficient to hypothesize on developmental specificities of pipefish features, such as on the absence of tooth primordia that could represent a very discrete and transient cell population. The ISH analyses would require a clean and precise spatiotemporal expression comparison of markers at the level of the entire pipefish and zebrafish specimens at well-defined stages, otherwise, the arguments proposed on teleost innovations and adaptations turn out to be very speculative.

We are appreciative of the reviewer’s feedback. We primarily used the *in situ* hybridization (ISH) data as supplementary to the scRNAseq library and we are aware that further evidence is necessary to identify origins of syngnathid’s evolutionary novelties. Our goal was to provide clues for the developmental genetic basis of syngnathid derived features. We hope that our study will inspire future investigations and are excited for the prospect that future research could include this reviewer’s ideas.

All of the developmental stages and species information for the embryos used were in the figure captions as well as in supplemental file 6. Because we primarily used wild caught embryos, we did not have specific ages of most embryos. Syngnathid species are challenging to culture in the laboratory, and extracting embryos requires euthanizing the father which makes it difficult to obtain enough embryos for ISH. In addition, embryos do not survive long when removed from the brood pouch prematurely. We supplemented our ISH with bay pipefish caught off the Oregon coast because these fish have large broods. Wild caught pregnant male bay pipefish were immediately euthanized, and their broods were fixed. Because we did not have their age, we classified them based on developmental markers such as presence of somites and the extent of craniofacial elongation. Although these classification methods are not ideal, they are consistent with the syngnathid literature (Sommer et al. 2012). Since the embryos used for the ISH were primarily wild caught, we had a few different developmental stages represented in our ISH data. For our tooth primordia search, we used embryos from the same brood (therefore, same stage) for these experiments.

We understand the concern for the degree of pigmentation in the samples. We completed numerous bleach trials before embarking on the *in situ* hybridization experiments. After completing a bleach trial with a probe created from the gene *tnmd* for ISH_,_ we noticed that the bleached embryos were missing expression domains found in the unbleached embryos. We were, therefore, concerned that using bleached embryos for our experiments would result incorrect conclusions about the expression domains of these genes. We sparingly used bleaching at older stages, hatched larvae, where it was fundamentally necessary to see staining. As stated above, the primary goal of this manuscript was to generate and annotate the first scRNA-seq atlas in a syngnathid, and the ISHs were utilized to support inferred cluster annotations only through a positive identification of marker gene expression in expected tissues/cells. Therefore, the obscuring of gene expression by pigmentation would have resulted in the absence of evidence for a possible cluster annotation, not an incorrect annotation.

For the ease of viewing the ISHs, we improved annotations and clarity. We increased the brightness and contrast of images. In the original submission, we had to lower the image resolution to make the submission file smaller. We hope that these improvements plus the true image quality improves clarity of ISH results. We also included alignments in our supplementary files of bay pipefish sequences to the Gulf pipefish probes to showcase the high degree of sequence similarity.

Sommer, S., Whittington, C. M., & Wilson, A. B. (2012). Standardised classification of pre-release development in male-brooding pipefish, seahorses, and seadragons (Family Syngnathidae). *BMC Developmental Biology*, *12*, 12–15.

To conclude, whereas the scRNA-seq dataset in this unconventional model organism will be useful for the community, the spatiotemporal and comparative expression analyses have to be thoroughly pushed forward to support the claims. Addressing these points is absolutely necessary to validate the data and to give new insights to understand the extraordinary evolution of the Syngnathidae family.

We really appreciate the reviewer’s enthusiasm for syngnathid research, and hope that the additional files and explanation of the supporting role of the ISHs have adequately addressed their concerns. We share the reviewer’s enthusiasm and are excited for future work that can extend this study.

**Reviewer #2 (Public Review):**
Summary:The authors present the first single-cell atlas for syngnathid fishes, providing a resource for future evolution & development studies in this group.Strengths:The concept here is simple and I find the manuscript to be well written. I like the in situ hybridization of marker genes - this is really nice. I also appreciate the gene co-expression analysis to identify modules of expression. There are no explicit hypotheses tested in the manuscript, but the discovery of these cell types should have value in this organism and in the determination of morphological novelties in seahorses and their relatives.

We are grateful for this reviewer’s appreciation of the huge amount of work that went into this study, and we agree that the in situ hybridizations (ISHs) support the scRNAseq study as we intended. We appreciate that the reviewer thinks that this work will add value to the syngnathid field.

Weaknesses:I think there are a few computational analyses that might improve the generality of the results.(1) The cell types: The authors use marker gene analysis and KEGG pathways to identify cell types. I'd suggest a tool like SAMap (https://elifesciences.org/articles/66747) which compares single-cell data sets from distinct organisms to identify 'homologous' cell types - I imagine the zebrafish developmental atlases could serve as a reasonable comparative reference.

We appreciate the reviewer’s request, and in fact we would have loved to integrate our dataset with zebrafish. However, syngnathid’s unique craniofacial development makes it challenging to determine the appropriate stage for comparison. While 3 days post fertilization (dpf) zebrafish data were appropriate for comparisons of certain cell types (e.g. epidermal cells), it would have been problematic for other cell types (e.g. osteoblasts) that are not easily detectable until older zebrafish stages. Therefore, determining equivalent stages between these species is difficult and contains potential for error. Future research should focus on trying to better match stages across syngnathids and zebrafish (and other fish species such as stickleback). Studies of this nature promise to uncover the role of heterochrony in the evo-devo of syngnathid’s unique snouts.

(2) Trajectory analyses: The authors suggest that their analyses might identify progenitor cell states and perhaps related differentiated states. They might explore cytoTRACE and/or pseudotime-based trajectory analyses to more fully delineate these ideas.

We thank the reviewer for this suggestion! We added a trajectory analysis using cytoTRACE to the manuscript. It complemented our KEGG analysis well (L172-175; S73) and has improved the manuscript.

(3) Cell-cell communication: I think it's very difficult to identify 'tooth primordium' cell types, because cell types won't be defined by an organ in this way. For instance, dental glia will cluster with other glia, and dental mesenchyme will likely cluster with other mesenchymal cell types. So the histology and ISH is most convincing in this regard. Having said this, given the known signaling interactions in the developing tooth (and in development generally) the authors might explore cell-cell communication analysis (e.g., CellChat) to identify cell types that may be interacting.

We agree! It would have been a wonderful addition to the paper to include a cell-cell communication analysis. One limitation of CellChat is that it only includes mouse and human orthologs. Given concerns of reviewer #3 for mouse-syngnathid comparisons, we decided to not pursue CellChat for this study. We are looking forward to future cell communication resources that include teleost fishes.

**Reviewer #3 (Public Review):**
Summary:This study established a single-cell RNA sequencing atlas of pipefish embryos. The results obtained identified unique gene expression patterns for pipefish-specific characteristics, such as fgf22 in the tip of the palatoquadrate and Meckel's cartilage, broadly informing the genetic mechanisms underlying morphological novelty in teleost fishes. The data obtained are unique and novel, potentially important in understanding fish diversity. Thus, I would enthusiastically support this manuscript if the authors improve it to generate stronger and more convincing conclusions than the current forms.

Thank you, we appreciate the reviewer’s enthusiasm!

Weaknesses:Regarding the expression of sfrp1a and bmp4 dorsal to the elongating ethmoid plate and surrounding the ceratohyal: are their expression patterns spatially extended or broader compared to the pipefish ancestor? Is there a much closer species available to compare gene expression patterns with pipefish? Did the authors consider using other species closely related to pipefish for ISH? Sfrp1a and bmp4 may be expressed in the same regions of much more closely related species without face elongation. I understand that embryos of such species are not always accessible, but it is also hard to argue responsible genes for a specific phenotype by only comparing gene expression patterns between distantly related species (e.g., pipefish vs. zebrafish). Due to the same reason, I would not directly compare/argue gene expression patterns between pipefish and mice, although I should admit that mice gene expression patterns are sometimes helpful to make a hypothesis of fish evolution. Alternatively, can the authors conduct ISH in other species of pipefish? If the expression patterns of sfrp1a and bmp4 are common among fishes with face elongation, the conclusion would become more solid. If these embryos are not available, is it possible to reduce the amount of Wnt and BMP signal using Crispr/Cas, MO, or chemical inhibitor? I do think that there are several ways to test the Wnt and/or BMP hypothesis in face elongation.

We appreciate the reviewer’s suggestion, and their recognition for challenges within this system. In response to this comment, we completed further *in situ* hybridization experiments in threespine stickleback, a short snouted fish that is much more closely related to syngnathids than is zebrafish, to make comparisons with pipefish craniofacial expression patterns (S76-S79). We added ISH data for the signaling genes (*fgf22*, *bmp4*, and *sfrp1a*) as well as *prdm16.* Through adding this additional ISH results, we speculated that craniofacial expression of *bmp4, sfrp1a,* and *prdm16* is conserved across species. However, compared to the specific ceratohyal/ethmoid staining seen in pipefish, stickleback had broad staining throughout the jaws and gills. These data suggest that pipefish have co-opted existing developmental gene networks in the development of their derived snouts. We added this interpretation to the results and discussion of the manuscript (L244-L248; L262-277; L444-470).

**Recommendations for the authors:**
**Reviewing Editor (Recommendations for the Authors)**:We hope that the eLife assessment, as well as the revisions specified here, prove helpful to you for further revisions of your manuscript.Revisions considered essential:(1) Marker genes and single-cell dataset analyses. While these analyses have been performed to a good standard in broad terms, there is a majority view here that cell type annotations and trajectory analyses can be improved. In particular, there is question about the choice of marker genes for the current annotation. For one it can depend on the use of single marker genes (see tnnti1 example for clusters 17 and 31). Here, we recommend incorporating results from SAMap and trajectory analysis (e.g., cytoTRACE or standard pseudotime).

Because of the reviewer comments, we became aware that we insufficiently communicated how cell clusters were annotated. We did mention in the manuscript that we did not use single marker genes to annotate clusters, but instead we used multiple marker genes for each cluster for the annotation process. We used both marker genes derived from our dataset and marker genes identified from zebrafish resources for cluster annotation. We chose single marker genes for each cluster for visualization purposes and for in situ hybridizations. However, it is clear from the reviewers’ comments that we needed to make more clear how the annotations were performed. To make this effort more clear in our revision, we included two new supplementary files – one with Seurat derived marker genes and one with marker genes derived from our DotPlot method. We also included extensive supplementary figures highlighting different markers. Using Daniocell, we identified 6 zebrafish markers per major cell type and showed their expression patterns in our atlas with FeaturePlots. We also included feature plots of the top 6 marker genes for each cluster. We hope that the addition of these 40+ plots (S3:S57) to the supplement fully addresses these concerns.

We appreciated the suggestion of cytotrace from reviewer #2! We ran cytotrace on three major cell lineages (neural, muscle, and connective; S73) which complemented our KEGG analysis in suggesting an undifferentiated fate for clusters 8, 10, and 16. We chose to not run SAMap because it is a scRNA-seq library integration tool. Although we compared our lectin epidermal findings to 3 dpf zebrafish scRNA-seq data, we did not integrate the datasets out of concern that we could draw erroneous conclusions for other cell types. Future work that explores this technical challenge may uncover the role of heterochrony in syngnathid craniofacial development. We detail these changes more fully in our responses to reviewers.

(2) The claims regarding evolutionary novelty and/or the genes involved are considered speculative. In part, this comes from relying too heavily on comparisons against zebrafish, as opposed to more closely related species. For example, the discussion regarding C-type lectin expression in the epidermis and KEGG enrichment (lines 358 - 364) seems confusing. Another good example here is the discussion on sfrp1a (lines 258 - 261). Here, the text seems to suggest craniofacial sfrp1a expression (or specifically ethmoid expression?) is connected to the development of the elongated snout in pipefish. However, craniofacial expression of sfrp1a is also reported in the arctic charr, which the authors grouped into fishes with derived craniofacial structures. Separately, sfrp2 expression was also reported in stickleback fish, for example. Do these different discussions truly support the notion that sfrp1a expression is all that unique in pipefish, rather than that pipefish and zebrafish are only distantly related and that sfrp1a was a marker gene first, and co-opted gene second? The authors should respond to the comments in the public review related to this aspect, and include more informative comparison and discussion.A much more nuanced discussion with appropriate comparisons and caveats would be strongly recommended here.

We appreciate this insight and used it as a motivator to complete and add select comparative ISH data to this manuscript. We added in situ hybridization experiments from stickleback fish for craniofacial development genes (sfrp_1a, prdm16, bmp4_, and *fgf22;* S76-S79). After adding stickleback ISH to the manuscript, we were able to make comparisons between pipefish and stickleback patterns and draw more informed conclusions (L244-L248; L262-277; L444-470). We added additional nuance to the discussion of the head, tooth (L485-489), and male pregnancy (L358-L391) sections to address concerns of study limitations. We describe in more detail these additional data in response to reviewers.

(3) In situ hybridization results: as already included above, there is generally weak labeling of species, developmental stages, and other markings that can provide context. The collective feeling here is that as it is currently presented, the ISH results do not go too far beyond simply illustrative purposes. To take these results further, more detailed comparison may be needed. At a minimum, far better labeling can help avoid making the wrong impression.

Based on the reviewers’ comments, we made changes to improve ISH clarity and add select comparative ISH findings. ISH was used to further interpretation of the scRNAseq atlas. All the developmental stages and species information for the embryos used were in the figure captions as well as in supplemental file 4. Since we primarily used wild caught embryos, we did not have specific ages of most embryos. The technical challenges of acquiring and staging Syngnathus embryos are detailed above. Because we did not have their age, we classified them based on developmental markers (such as presence of somites and the extent of craniofacial elongation). Although these classification methods are not ideal, they are consistent with the syngnathid literature (Sommer et al. 2012).

We followed reviewer #1’s recommendations by adding an annotated graphic of a pipefish head, aligning bay and Gulf pipefish sequences for the probe regions, expanding out our supplemental figures for ISH into a figure for each probe, and improving labeling. These changes improved the description of the ISH experiments and have increased the quality of the manuscript.

We would have loved to complete detailed comparative studies as suggested, but doing such a complete analysis was not feasible for this study. Therefore, we completed an additional focused analysis. We followed reviewer #3’s idea and added ISHs from threespine stickleback, a short snouted fish, for 4 genes (*sfrp1a, prdm16, fgf22,* and *bmp4*). While more extensive ISHs tracking all marker genes through a variety of developmental stages in pipefish and stickleback would have provided crucial insights, we feel that it is beyond the scope of this study and would require a significant amount of additional work. We, thus, primarily interpreted the ISH results as illustrative data points in our discussion. As we state in the response to reviewer 1, the generation and annotation of the first scRNA-seq atlas in a syngnathid is the primary goal of this manuscript. The ISHs were utilized primarily to support inferred cluster annotations if a positive identification of marker gene expression in expected tissues/cells occurred.

**Reviewer #1 (Recommendations For The Authors):**
While the scRNA-seq dataset offers a valuable resource for evo-devo analyses in fish and the hypotheses are of interest, critical aspects should be strengthened to support the claims of the study.Concerning the scRNA-seq dataset, the major points to be addressed are listed below:- Supplementary file 3 reports the single markers used to validate cluster annotations. To confirm cluster identities, more markers specific to each cluster should be highlighted and presented on the UMAP.

We recognize the reviewer’s concern and had in reality used numerous markers to annotate the clusters. Based upon the reviewer’s comment we decided to make this clear by creating feature plots for every cluster with the top 6 marker genes. These plots showcase gene specificity in UMAP space. We also added feature plots for zebrafish marker genes for key cell types. Through these changes and the addition of 54 supplementary figures (S3:S57), we hope that it is clear that numerous markers validated cluster identity.

For example, as clusters 17 and 37 share the same tnnti1 marker, which other markers permit to differentiate their respective identity.

This is a fair point. Cluster 17 and 37 both are marked by a *tnni1* ortholog.

Different paralogous co-orthologs mark each cluster (cluster 17: *LOC125989146*; cluster 37: *LOC125970863).* In our revision to the above comment, additional (6) markers per cluster were highlighted which should remedy this concern.

- L146: the low number of identified cartilaginous cells (only 2% of total connective tissue cells) appears aberrant compared to bone cell number, while Figure 1 presents a welldeveloped cartilaginous skeleton with poor or no signs of ossification. Please discuss this point.

We also found this to be interesting and added a brief discussion on this subject to the results section (L147-L149). Single cell dissociations can have variable success for certain cell types. It is possible that the cartilaginous cells were more difficult to dissociate than the osteoblast cells.

- L162: pax3a/b are not specific to muscle progenitors as the genes are also expressed in the neural tube and neural crest derivatives during organogenesis. Please confirm cluster 10 identity.

Thank you for the reminder, we added numerous feature plots that explored zebrafish (from Daniocell) and pipefish markers (identified in our dataset). Examining zebrafish satellite muscle markers (*myog, pabpc4,* and *jam2a*) shows a strong correspondence with cluster #10.

- L198: please specify in the text the pigment cell cluster number.

We completed this change.

- L199: it is not clear why considering module 38 correlated to cluster 20 while modules 2/24 appear more correlated according to the p-value color code.

We thank the reviewer for pointing this confusing element out! Although the t-statistic value for module 38 (3.75) is lower than the t-statistics for modules 2 and 24 (5.6 and 5.2, respectively), we chose to highlight module 38 for its ‘connectivity dependence’ score. In our connectivity test, we examined whether removing cells from a specific cell cluster reduced the connectivity of a gene network. We found that removing cluster 20 led to a decrease in module 38’s connectivity (-.13, p=0) while it led to an increase in modules 2 and 24’s connectivity (.145, p=1; .145, p=9.14; our original supplemental files 9-10). Therefore, the connectivity analysis showed that module 38’s structure was more dependent on cluster 20 than in comparison with modules 2 and 24. Although you highlighted an interesting quandary, we decided that this is tangential to the paper and did not add this discussion to the manuscript.

- Please describe in the text Figure 4A.

Completed, we thank the reviewer for catching this!

Concerning embryo stainings, the major points to be addressed are listed below:- Figure 1: please enhance the light/contrast of figures to highlight or show the absence of alcian/alizarin staining. Mineralized structures are hardly detectable in the head and slight differences can be seen between the two samples. The developmental stage should be added. Please homogenize the scale bar format (remove the unit on panels E and, G as the information is already in the text legend). It would be useful to illustrate the data with a schematic view of the structures presented in panels B, and E, and please annotate structures in the other panels.

We thank the reviewer for these suggestions to improve our figure. We increased the brightness and contrast for all our images. We also added an illustration of the head with labels of elements. As discussed, we used wild caught pregnant males and, therefore, do not know the exact age of the specimens. However, we described the developmental stage based on morphological observations. Slight differences in morphology between samples is expected. We and others have noticed that

developmental rate varies, even within the same brood pouch, for syngnathid embryos. We observed several mineralization zones including in the embryos including the upper and lower jaws, the mes(ethmoid), and the pectoral fin. We recognize the cartilage staining is more apparent than the bone staining, though increasing image brightness and contrast did improve the visibility of the mineralization front.

- All ISH stainings and images presented in Figures 4-6/ Figures S2-3 should be revised according to comments provided in the public review.

We thank the reviewer for providing thorough comments, we provided an in-depth response to the public review. We made several improvements to the manuscript to address their concerns.

- Figure 4: Figure 4B should be described before 4C in the text or inverse panels / L222 the Meckel's cartilage is not shown on Figure 4C. The schematic views in H should be annotated and the color code described / the ISH data must be completed to correlate spatially clusters to head structures.

We thank the reviewer for pointing this out, we fixed the issues with this figure and added annotations to the head schematics.

- Figure 5: typo on panels 'alician' = alcian.

We completed this change.

- Figures S2-3: data must be better presented, polished / typo in captions 'relavant' = relevant.

Thank you for this critique, we created new supplementary figures to enhance interpretation of the data (S59-S71). In these new figures, we included a feature plot for each gene and respective ISHs.

- Figure S3: soat2 = no evidence of muscle marker neither by ISH presented nor in the literature.

We realized this staining was not clear with the previous S2/S3 figures. Our new changes in these supplementary figures based on the reviewer’s ideas made these ISH results clearer. We observed soat2 staining in the sternohyoideus muscle (panel B in S71).

Other points:- The cartilage/bone developmental state (Alcian/alizarin staining) and/or ISH for classical markers of muscle development (such as pax3/myf5) could be used to clarify the This could permit the completion of a comparative analysis between the two species and the interpretation of novel and adaptative characters.

We appreciate this idea! We thought deeply about a well characterized comparative analysis between pipefish and zebrafish for this study. We discussed our concerns in our public response to reviewer 2. We found that it was challenging to stage match all cell types, and were concerned that we could make erroneous conclusions. For example, our pipefish samples were still inside the male brood pouch and possessed yolk sacs. However, we found osteoblast cells in our scRNAseq atlas, and in alizarin staining. Although zebrafish literature notes that the first zebrafish bone appears at 3 dpf (Kimmel et al. 1995), osteoblasts were not recognized until 5 dpf in two scRNAseq datasets (Fabian et al. 2022; Lange et al. 2023). A 5dpf zebrafish is considered larval and has begun hunting. Therefore, we chose to not integrate our data out of concern that osteoblast development may occur at different timelines between the fishes.

Fabian, P., Tseng, K.-C., Thiruppathy, M., Arata, C., Chen, H.-J., Smeeton, J., Nelson, N., & Crump, J. G. (2022). Lifelong single-cell profiling of cranial neural crest diversification in zebrafish. *Nature Communications 2022 13:1*, *13*(1), 1–13.

Lange, M., Granados, A., VijayKumar, S., Bragantini, J., Ancheta, S., Santhosh, S., Borja, M., Kobayashi, H., McGeever, E., Solak, A. C., Yang, B., Zhao, X., Liu, Y., Detweiler, A. M., Paul,

S., Mekonen, H., Lao, T., Banks, R., Kim, Y.-J., … Royer, L. A. (2023). Zebrahub – Multimodal Zebrafish Developmental Atlas Reveals the State-Transition Dynamics of Late-Vertebrate Pluripotent Axial Progenitors. *BioRxiv*, 2023.03.06.531398.

Kimmel, C., Ballard, S., Kimmel, S., Ullmann, B., Schilling, T. (1995). Stages of Embryonic Development of the Zebrafish. Developmental Dynamics 203:253:-310.

'in situs' in the text should be replaced by 'in situ experiments'.

We made this change (L395, L663, L666, L762).

- Lines 562-565: information on samples should be added at the start of the result section to better apprehend the following scRNA-seq data.

We thank the reviewer for pointing out this issue. Although we had a few sentences on the samples in the first paragraph of the result section, we understand that it was missing some critical pieces of information. Therefore, we added these additional details to the beginning of the results section (L126-L132).

- Lines 629-665: PCR with primers designed on gulf pipefish genome could be performed in parallel on bay and gulf cDNA libraries, and amplification products could be sequenced to analyze alignment and validate the use of gulf pipefish ISH probes in bay pipefish embryos. Probe production could also be performed using gulf primers on bay pipefish cDNA pools.

After the submission of this manuscript, a bay pipefish genome was prepared by our laboratory. We used this genome to align our probes, these alignments demonstrate strong sequence conservation between the species. We included these alignments in our supplemental files.

- L663: the bleaching step must be optimized on pipefish embryos.

We understand this concern and had completed several bleach optimization experiments prior to publication. Although we found that bleaching improved visibility of staining, we noticed with the probe *tnmd* that bleached embryos did not have complete staining of tendons and ligaments. The unbleached embryos had more extensive staining than the bleached embryos. We were concerned that bleaching would lead to failures to detect expression domains (false negatives) important for our analysis. Therefore, we did not use bleaching with our in situs experiments (except with hatched fish with a high degree of pigmentation).

- Indicate the number of specimens analyzed for each labeling condition.

We thank the reviewer for noticing this issue. We added this information to the methods (L766-767).

- Describe the fixation and pre-treatment methods previous to ISH and skeleton stainings

We thank the reviewer for pointing out this issue, we added these descriptions (L765-766; L772-774).

**Reviewer #3 (Recommendations For The Authors):**
(1) If sfrp1a expression is observed also in other fish species with derived craniofacial structures, it's important to discuss this more in the Discussion. This could be a common mechanism to modify craniofacial structures, although functional tests are ultimately required (but not in this paper, for sure). Can lines 421-428 involve the statement "a prolonged period of chondrocyte differentiation" underlies craniofacial diversity?

This is a great idea, and we added a sentence that captures this ethos (L451-452).

(2) Lines 334-346 need to be rephrased. It's hard to understand which genes are expressed or not in pipefish and zebrafish. Did "23 endocytosis genes" show significant enrichment in zebrafish epidermis, or are they expressed in zebrafish epidermis?

We thank the reviewer for this comment, we re-phrased this section for clarity (L365-368).

(3) Figure 4 is missing the "D" panel and two "E" panels.

We thank the reviewer for noticing this, we fixed this figure.

(4) Line 302: "whole-mount" or "whole mount"

We thank the reviewer for the catch!